# Enhanced plasma half-life and efficacy of engineered human albumin-fused GLP-1 despite enzymatic cleavage of its C-terminal end

Jeannette Nilsen[1,2,3,7], Kristin Hovden Aaen [1,2,3,7], Sopisa Benjakul [1,2,3,7], Fulgencio Ruso-Julve [1,2,3], Thomas Uwe Greiner [4], Daniela Bejan[1,2,3], Maria Stensland[1], Sachin Singh[1], Tilman Schlothauer [5], Inger Sandlie[6] & Jan Terje Andersen [1,2,3] ✉

Albumin has a long plasma half-life due to engagement of the neonatal Fc receptor (FcRn), which prevents intracellular degradation. However, its C-terminal end can be cleaved by carboxypeptidase A, and removal of the last leucine residue (L585) weakens receptor binding, reducing its half-life from 20 days to 3.5 days in humans. This biology has so far been overlooked when designing human albumin-fused biologics. Thus, there is a need for an engineering strategy to secure favorable FcRn binding and pharmacokinetic properties. Here, we show that a branched aliphatic amino acid or methionine at position 585 of albumin is required for optimal receptor binding, which cannot be replaced to prevent enzymatic cleavage without negatively affecting FcRn engagement. As a solution, we report that C-terminally cleaved albumin can be efficiently rescued from intracellular degradation by introducing amino acid substitutions that improve FcRn binding. This albumin-engineering strategy was also effective when applied with a therapeutic fusion partner, glucagon-like peptide 1 (GLP-1), resulting in a 2-fold increase in plasma half-life and prolonged efficacy in human FcRn transgenic mice. We demonstrate how human albumin fusions should be tailored to ensure a long plasma half-life and enhanced efficacy of fused biologics, despite potential C-terminal cleavage in vivo.

Albumin (66.5 kDa) is a plasma protein expressed by hepatocytes of the liver, and functions as a transporter of free fatty acids, thyroid hormones, and many other insoluble molecules[1]. In humans, albumin is secreted as a single polypeptide chain of 585 amino acids that folds into three homologous domains (DI, DII, and DIII), forming a heart-shaped structure[2–4]. Albumin is the most abundant protein in the blood, as it constitutes more than 60% of the total plasma protein pool, and therefore, plays a vital role in sustaining oncotic pressure[1]. In addition, albumin has an average plasma half-life of three weeks in humans[1,5], a feature that is increasingly exploited to improve the pharmacokinetic (PK) properties of short-lived biologics, including peptide hormones,

cytokines, coagulation factors, and antibody fragments[6–15]. Such protein-based biologics can be genetically fused to either the N-terminal or C-terminal end of albumin, or to both[6–15].

One such example is glucagon-like peptide 1 (GLP-1) (3.3 kDa), a small peptide hormone with a plasma half-life of only two minutes due to enzymatic degradation by dipeptidyl peptidase-4 (DDP-4) in the blood and rapid excretion through the kidneys[16–19]. GLP-1 is secreted by intestinal L-cells in response to food intake and promotes insulin secretion from pancreatic β-cells, thereby playing a key role in regulating the blood glucose concentration[20–23]. However, the therapeutic use of the native peptide is restricted by its short plasma half-life. Thus, recombinant long-acting

¹Department of Immunology, Oslo University Hospital Rikshospitalet, N-0372 Oslo, Norway. ²Department of Pharmacology, Institute of Clinical Medicine, University of Oslo, N-0372 Oslo, Norway. ³Precision Immunotherapy Alliance (PRIMA), University of Oslo, N-0372 Oslo, Norway. ⁴Wallenberg Laboratory, Department of Molecular and Clinical Medicine, Institute of Medicine, University of Gothenburg, 413 45 Gothenburg, Sweden. ⁵Roche Pharma Research and Early Development (pRED), Therapeutic Modalities, Roche Innovation Center Munich, Roche Diagnostics GmbH, 82377 Penzberg, Germany. ⁶Department of Biosciences, University of Oslo, N-0371 Oslo, Norway. ⁷These authors contributed equally: Jeannette Nilsen, Kristin Hovden Aaen, Sopisa Benjakul. ✉e-mail: j.t.andersen@medisin.uio.no

GLP-1 receptor agonists (RAs) have been developed for the treatment of type 2 diabetes *mellitus* (T2DM)[24,25], and were also recently approved to treat obesity[26,27]. For instance, albiglutide (73.0 kDa) is composed of two copies of a modified GLP-1 (residues 7–36) fused in tandem to the N-terminal end of wild-type (WT) human albumin[10,28]. The fusion product is resistant to DPP-4 degradation, due to an alanine to glycine substitution at position 2 (A2G) in each of the peptide sequences[28,29], and has a plasma half-life of five days on average in humans[30–32]. Albiglutide was approved for once-weekly treatment of T2DM, and marketed as Tanzeum/Eperzan from 2014 to 2018 by GlaxoSmithKline[32], before the company discontinued the product due to financial priorities in a highly competitive market.

The neonatal Fc receptor (FcRn) plays a key role in regulating albumin homeostasis[33], and therefore, the FcRn-albumin relationship should be considered when designing albumin-based fusion proteins. The receptor is a major histocompatibility complex (MHC) class I-related molecule, which binds and regulates the plasma half-life of albumin as well as the structurally unrelated immunoglobulin G (IgG) antibody[34–36]. FcRn is broadly expressed across a range of hematopoietic and non-hematopoietic cells[37–40], including endothelial cells, where it is predominantly localized within intracellular endosomal compartments[41–49]. Albumin and IgG can be taken up by endothelial cells lining blood vessels by fluid-phase pinocytosis and enter the endosomal pathway. In contrast to other proteins that do not engage FcRn, the two ligands are prevented from progressing to the lysosomes for degradation by binding to the receptor in early sorting endosomes[41,43,47,49,50]. FcRn-ligand complexes are instead sorted into vesicles that traffic back to the plasma membrane and undergo exocytosis[42–46,49]. Crucial for this recycling mechanism is the pH-dependence of the interaction between FcRn and its ligands, where binding only occurs at mildly acidic conditions (pH < 6.5) and not at physiological pH (pH 7.4)[33,51,52]. This allows association of the ligands within the acidic endosomes and dissociation upon exposure to the physiological pH of the blood. Consequently, FcRn rescues both albumin and IgG from intracellular lysosomal degradation, which explains their long plasma half-lives and high abundance in plasma.

FcRn interacts with albumin and IgG at distinct binding sites on its heterodimeric structure, which consists of a transmembrane MHC class I-like heavy chain (HC) and a soluble, non-covalently associated β2-microglobulin (β2m) subunit[52–56]. The principal receptor binding site of albumin is located within its C-terminal DIII[52,57–59]. Importantly, an intact C-terminal end of albumin is required for optimal binding to FcRn, and we recently demonstrated that truncation of the last leucine residue at position 585 (L585) reduced the binding affinity by 5-fold[60]. Further investigation of the protein dynamics suggested that L585 contributes to the flexibility of DIII, which undergoes large conformational changes upon FcRn engagement[52,59,60]. Notably, a significant portion of albumin in the blood lacks the C-terminal L585 residue, ranging from 3 to 15% in healthy individuals and up to 93% in patients with pancreatitis or other pancreatic disorders[60–62]. Studies support that carboxypeptidase A (CPA), a pancreas-derived exo-protease with specificity for C-terminal aromatic and aliphatic amino acid residues, is responsible for the removal of L585[60,63–65]. Enzymatic cleavage of albumin results in less efficient FcRn-mediated rescue from lysosomal degradation and, consequently, a dramatically shorter plasma half-life of 3.5 days in humans[60]. As GLP-1 in albiglutide requires an available N-terminal end for engagement with the GLP-1 receptor (GLP-1R)[66,67], albumin is fused to the C-terminal end of the peptide. However, such design leaves the C-terminal end of albumin free and susceptible to enzymatic cleavage by CPA.

In this study, we demonstrate that human albumin requires a branched or medium-long aliphatic amino acid in the C-terminal end at position 585 for optimal binding to human FcRn, and that the L585 residue can not be substituted with an amino acid unrecognizable by CPA without compromising receptor binding. Instead, we took advantage of three amino acid substitutions (E505Q, T527M, and K573P; QMP) in albumin reported to improve its pH-dependent binding to FcRn[11], and found that this albumin engineering compensated for the decrease in receptor binding caused by L585 cleavage. When administered by intravenous (IV) injections, the

QMP-engineered albumin variant lacking L585 demonstrated enhanced cellular recycling and extended plasma half-life in transgenic mice expressing human FcRn, comparable to that of full-length QMP. Furthermore, while enzymatic removal of the L585 residue decreased FcRn binding for fusion designs with WT albumin, this was not the case for the corresponding QMP-engineered variants. In turn, the QMP substitutions improved the PK properties of a recombinant version of the GLP-1 albumin fusion albiglutide, irrespective of the presence of L585. Consequently, the QMP-engineered GLP-1 albumin fusion demonstrated prolonged glucose-lowering capacity in vivo, compared to the WT counterpart. Thus, we propose QMP engineering as a strategy to ensure a long plasma half-life and enhanced therapeutic efficacy of albumin fusion proteins despite C-terminal cleavage.

## Results

### L585 substitution of albumin to a non-aliphatic amino acid weakens FcRn binding

The C-terminal DIII of human albumin harbors the primary binding site for human FcRn (Fig. 1a)[52,57–59], and truncation of a single C-terminal amino acid residue (L585) weakens FcRn binding affinity and, consequently, reduces its in vivo plasma half-life[60,65]. While CPA catalyzes hydrolysis of the amide bond between L585 and G584, it does not of the following amide bond between G584 and L583 (Fig. 1b)[60,63,65]. This is in line with the reported specificity of CPA, which is C-terminal amino acid residues with branched aliphatic or aromatic side chains[64].

To investigate whether albumin could be engineered to evade L585 cleavage and the resulting reduction in FcRn binding, we constructed a panel of 18 full-length 585-engineered human albumin variants, in which the C-terminal leucine residue was substituted with each of the other naturally occurring amino acids, except for cysteine (Fig. 1b). The human albumin variants were expressed in a serum-free transient Expi293 expression system, and all were secreted at comparable quantities to that of WT albumin (25–46 mg/L) (Supplementary Fig. 1a). The proteins were purified from cell culture supernatant using a human albumin affinity column, followed by size exclusion chromatography (SEC), which yielded monomeric fractions that migrated on non-reducing sodium dodecyl-sulfate polyacrylamide gel electrophoresis (SDS-PAGE) gels according to their molecular weight (MW) (66.5 kDa) (Fig. 1c).

To assess how the L585 substitutions in human albumin affect FcRn binding, we performed surface plasmon resonance (SPR) experiments, where biotinylated human FcRn was captured on a streptavidin (SA)-coupled sensor chip before a single concentration of each albumin variant was injected over at pH 5.5 (Fig. 1d). The SPR sensorgrams showed that 15 of the 18 amino acid substitutions accelerated dissociation from the receptor, and thus weakened the binding to FcRn (Fig. 1e–h). Only the albumin variants with an isoleucine (L585I), a methionine (L585M), or a valine (L585V) showed binding curves similar to that obtained for the WT albumin (Fig. 1h, i). Injections of concentration series of these three albumin variants confirmed that they bound FcRn with similar kinetics and affinity to that of WT albumin (Table 1, Supplementary Fig. 2). However, in line with the specificity of CPA, the C-terminal residues of the L585I, L585M, and L585V variants were cleaved in vitro upon exposure to the enzyme for 5 hours at 37 °C, as confirmed by liquid chromatography with tandem mass spectrometry (LC-MS/MS) analysis of the C-terminal peptides (Supplementary Table 1). These data support that the presence of an amino acid residue with a branched aliphatic side chain, or a methionine, at position 585 of albumin is required for optimal FcRn binding. Thus, a human albumin variant resistant to enzymatic cleavage by CPA could not be obtained through L585 engineering without compromising FcRn binding.

### QMP substitutions compensate for decreased FcRn binding caused by L585 cleavage

The three amino acid substitutions introduced in DIII of human albumin, QMP, greatly improve its pH-dependent binding to human FcRn

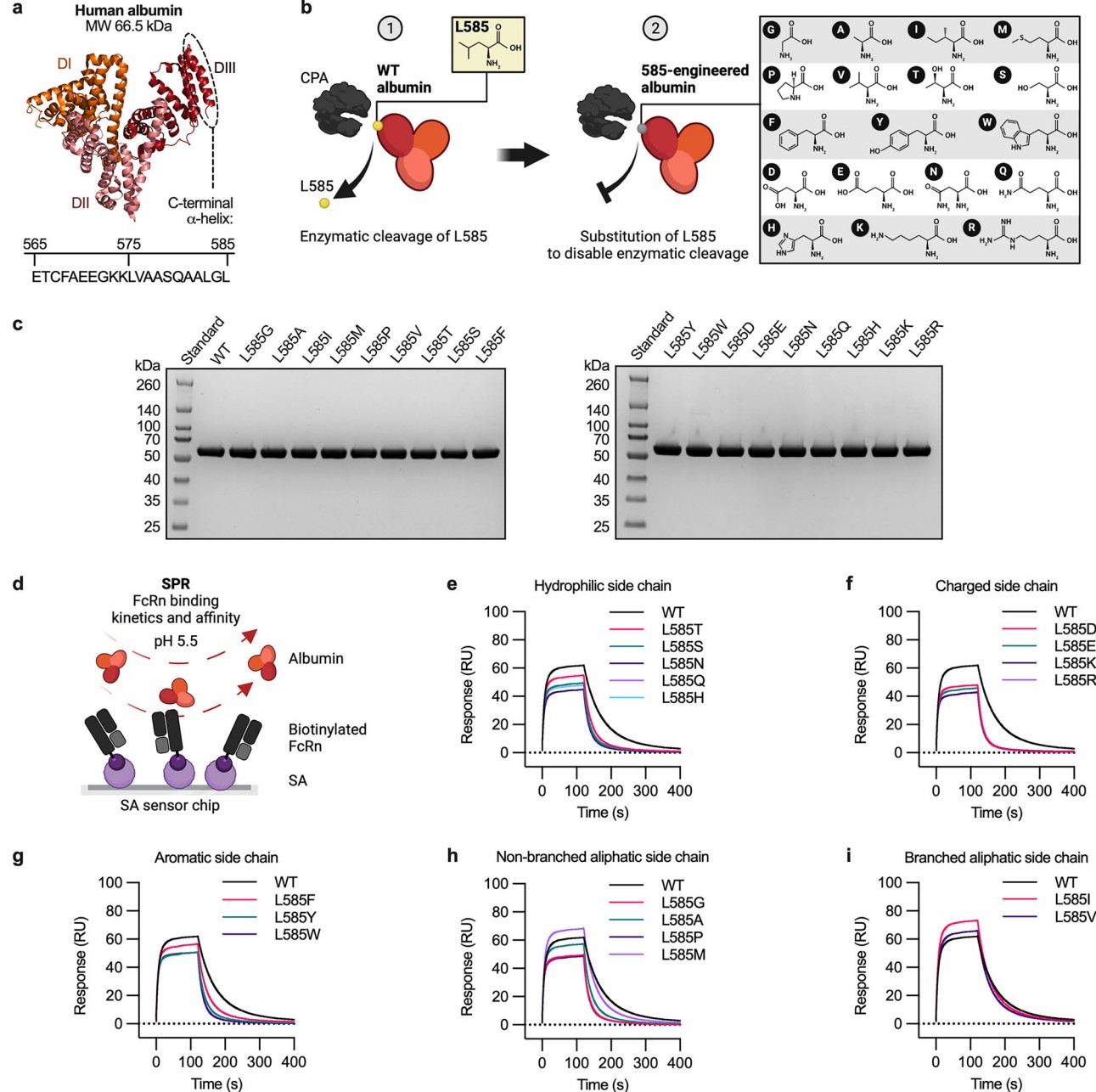

**Fig. 1 | Albumin lacking a C-terminal branched or medium-long aliphatic residue at position 585 shows reduced binding to FcRn. a** Illustration of a solved crystal structure of human albumin (66.5 kDa), with DI, DII, and DIII shown in orange, salmon pink, and ruby, respectively. The C-terminal α-helix (residues 565–585) of DIII is indicated with its corresponding amino acid sequence. The illustration was made using PyMOL with the solved crystal structure of human albumin (PDB ID: 6WUW)[95]. **b** Illustration showing (1) enzymatic cleavage of the C-terminal end of human WT albumin by CPA, and (2) engineering of human albumin in position 585 to disable enzymatic cleavage. **c** Non-reducing SDS-PAGE gels showing purified fractions of recombinant full-length human WT albumin and 585-engineered albumin variants (66.5 kDa) produced in Expi293F cells. **d** Illustration of the SPR setup used to measure FcRn–albumin binding kinetics and affinity at pH 5.5, in which biotinylated human FcRn (purple-black/grey) was immobilized on an SA sensor chip, followed by injection of human albumin (red). **e–i** Representative SPR sensorgrams showing the binding a single concentration (500 nM) of WT albumin and albumin variants, having an amino acid residue at position 585 with **e** hydrophilic, **f** charged, **g** aromatic, **h** non-branched aliphatic, or **i** branched aliphatic side chain, to immobilized FcRn at pH 5.5. Illustrations in **a**, **b**, and **d** were created with BioRender[100].

(Fig. 2a)[9,11,13–15]. To investigate whether QMP engineering would compensate for the decreased receptor binding caused by L585 cleavage, we first tested if CPA would cleave the C-terminal end of the QMP (66.5 kDa) and also included WT albumin as a control (Fig. 2b, Supplementary Fig. 1b). WT albumin and QMP were incubated for 5 h at 37 °C, with and without CPA, followed by an LC-MS/MS analysis. Upon exposure to CPA, the C-terminal leucine residue of both WT albumin and QMP was removed, whereas the samples without the enzyme remained intact (Supplementary Table 2).

Thus, as for WT albumin, the C-terminal end of QMP is susceptible to enzymatic cleavage by CPA.

To evaluate whether L585 cleavage would affect the pH-dependent FcRn binding properties of QMP, an enzyme-linked immunosorbent assay (ELISA) was performed (Fig. 2c). Briefly, an Fc-engineered human IgG1 mutant variant (M252Y/S254T/T256E/H433K/N434F; YTE/KF), which binds strongly to the receptor at both acidic and physiological pH[68], was coated in wells, before His-tagged human FcRn was captured via its IgG

binding site. Then, titrated amounts of WT albumin and QMP, either not exposed or exposed to CPA, were added and detected with an alkaline phosphatase (ALP)-conjugated anti-human albumin antibody. While WT albumin showed slightly weaker binding to FcRn at pH 5.5 upon enzymatic cleavage, L585-cleaved QMP bound to the receptor as strongly as full-length QMP (Fig. 2d). In agreement, the QMP variant recombinantly expressed without the L585 residue (QMP/LX) (66.5 kDa) (Fig. 2b, Supplementary Fig. 1b) bound equally well to FcRn as full-length QMP at pH 5.5 (Fig. 2e). Moreover, as for full-length WT albumin and QMP, no receptor binding was detected for QMP/LX at pH 7.4 (Fig. 2f).

To determine the binding kinetics and affinity, we performed SPR experiments, where concentration series of the QMP-engineered albumin variants were injected over immobilized human FcRn at pH 5.5 (Fig. 1d). In line with previous reports, QMP (0.3 nM) showed greater binding affinity to the receptor WT albumin (635.6 nM) (Fig. 2g, h, Table 1). QMP/LX (0.9 nM) exhibited a 3-fold weaker binding affinity compared to full-length QMP, due to a slightly slower association rate and a faster dissociation rate (Fig. 2i, Table 1). Similar to WT albumin, both QMP variants demonstrated strict pH-dependent binding to FcRn, with rapid dissociation from the immobilized receptor upon injection of a pH 7.4 buffer (Supplementary Fig. 3a–c). Thus, introduction of the QMP substitutions in albumin compensates for the decrease in FcRn binding caused by L585 cleavage, without disrupting the pH-dependency of the interaction. Moreover, nano differential scanning fluorimetry (nanoDSF) demonstrated similar melting temperatures ($T_m$) for both QMP (63.1 °C) and QMP/LX (64.2 °C), comparable to that of WT albumin (63.7 °C), supporting that the thermal stability of QMP is unaffected by the absence of L585 (Table 2, Supplementary Fig. 4a, b). Lastly, the theoretical net charge of the proteins was determined as a function of pH (pH 1.0–13.9) (Supplementary Fig. 5a). An isoelectric point (pI) of 5.8, and net charges of +2.8 and −14.7 at pH 5.5 and pH 7.4, respectively, were calculated for the albumin variants (Supplementary Table 3). Thus, QMP provides both favorable FcRn binding properties and a stable albumin protein, irrespective of the presence of L585.

## QMP enhances cellular recycling and plasma half-life despite the absence of L585

To study the importance of L585 in a cellular context, we performed a human endothelial cell-based recycling assay (HERA)[48] using human microvascular endothelial cells (HMEC-1) that overexpress human FcRn (HMEC-1-FcRn) (Fig. 3a)[69]. Briefly, adherent HMEC-1-FcRn cells were grown to confluence before a 1-h starvation period. Next, equimolar amounts of full-length and truncated QMP were added and incubated for 3 h to allow for cellular uptake. The cells were then lysed to quantify the amounts of albumin inside, or washed and further incubated in serum-free medium for an additional 3 h to allow for release of recycled albumin back into the medium. The amounts of albumin inside the lysed cells, or recycled albumin in the medium, were quantified by ELISA, in which albumin was captured on an anti-human albumin antibody and detected with an ALP-conjugated anti-human albumin antibody. The results showed that about 5-fold more of both QMP and QMP/LX than WT albumin was taken up (Supplementary Fig. 6a, c). Moreover, QMP and QMP/LX were recycled equally well, about 8 to 10-fold more efficiently than WT albumin (Fig. 3b, Supplementary Fig. 6e).

To investigate whether the truncated QMP/LX would be recycled as well as the full-length QMP in vivo, we performed a half-life study in transgenic Tg32 mice expressing human FcRn (human FcRn Tg32) (Fig. 3c). The use of humanized mice was required, as it is well-established that human albumin binds poorly to mouse FcRn[13,70–73]. The albumin variants were administered by IV injections into the lateral tail vein, followed by blood sampling for up to 23 days (Fig. 3c). The protein concentrations in plasma were determined by the human albumin-specific ELISA. One day after injection, the amount of QMP in the plasma was twice that of WT albumin, a difference that increased to more than 4-fold on day 10 (Fig. 3d). As previously reported[11], the plasma half-life of QMP (4.5 days) was twice as long as that of WT albumin (2.3 days) (Fig. 3e). To gain further insight, a non-compartmental PK analysis was performed using gPKPDSim[74], which revealed a 3.8-fold slower clearance rate (CL) and a 2.4-fold longer mean residence time (MRT) of QMP compared with WT albumin (Supplementary Table 4). Moreover, only minor differences in the plasma concentration of QMP and QMP/LX were detected over time (Fig. 3d, e). Consequently, QMP/LX exhibited a long plasma half-life despite lacking L585 (4.3 days) (Fig. 3e, Supplementary Table 4). Thus, irrespective of the presence of the C-terminal L585 residue, QMP successfully competes for binding to human FcRn in vivo, resulting in an extended plasma half-life.

## Design of engineered GLP-1 albumin fusions with favorable FcRn binding properties

The FDA-approved GLP-1 RA, Tanzeum (albiglutide), consists of two repeats of the DDP-4-resistant GLP-1 (residues 7–36) peptide fused in tandem to the N-terminal end of human albumin (Fig. 4a)[10,28]. Consequently, exposure to CPA leads to enzymatic cleavage of the C-terminal end of albumin, reducing its binding to FcRn, as previously reported[60]. To investigate whether QMP would secure a long plasma half-life of a GLP-1 albumin fusion, we constructed a recombinant fusion protein with the identical amino acid sequence as Tanzeum (albiglutide-mimic: GLP-1 WT), as well as a modified version containing the QMP substitutions (GLP-1 QMP) (Fig. 4a). The two fusion proteins were designed with either an intact C-terminal end (GLP-1 WT and GLP-1 QMP) or a truncated end lacking L585 (GLP-1 LX and GLP-1 QMP/LX) (Fig. 4a). The four constructs were transiently expressed using the Expi293 expression system, which resulted in high yields of purified monomeric protein fractions (180–513 mg/L) that migrated according to their MW (73 kDa), and at the same rate as Tanzeum, on an SDS-PAGE gel, both under non-reducing and reducing conditions (Fig. 4b, Supplementary Fig. 1c). Moreover, the nanoDSF-derived $T_m$ values obtained for the designed GLP-1 albumin fusions (56.4–57.4 °C) were comparable to that of Tanzeum (58.8 °C) (Table 2, Supplementary Fig. 4c, d). A theoretical pI of 5.8, and net charges of +2.9 and −15.5 at pH 5.5 and pH 7.4, respectively, were calculated for the GLP-1 albumin fusions (Supplementary Fig. 5b, Supplementary Table 3).

As reported for Tanzeum[60], the C-terminal L585 of the GLP-1 WT albiglutide-mimic and GLP-1 QMP fusions was removed upon exposure to CPA (Supplementary Table 2). After exposure to CPA, GLP-1 WT showed

**Table 1 | SPR-derived FcRn binding kinetic constants for albumin variants and GLP-1 albumin fusions**

| Variant[a] | $k_a$ ($10^4$ M$^{-1}$ s$^{-1}$)[b] | $k_d$ ($10^{-4}$ s$^{-1}$)[b] | $K_D$ (nM)[b] | $\chi^2$ (RU$^2$)[c] |
|---|---|---|---|---|
| WT | 2.7 | 171.6 | 635.6 | 0.6 |
| L585I | 2.9 | 175.2 | 604.1 | 1.5 |
| L585M | 2.9 | 196.6 | 677.9 | 0.7 |
| L585V | 3.4 | 232.7 | 684.4 | 0.4 |
| QMP | 14.4 | 0.4 | 0.3 | 0.01 |
| QMP/LX | 11.5 | 1.0 | 0.9 | 0.01 |
| Tanzeum | 15.7 | 119.4 | 76.1 | 0.3 |
| GLP-1 WT | 7.3 | 83.2 | 114.0 | 0.8 |
| GLP-1 LX | 4.3 | 221.4 | 514.9 | 0.8 |
| GLP-1 QMP | 38.1 | 0.1[d] | 0.03 | 0.1 |
| GLP-1 QMP/LX | 32.6 | 0.5 | 0.2 | 1.5 |

[a]Biotinylated human FcRn (~100 RU) was immobilized on an SA sensor chip, and serial dilutions of the human albumin variants or GLP-1 albumin fusions were injected at pH 5.5 (n = 2).
[b]A simple first-order (1:1) Langmuir interaction model was used to derive the kinetic constants, where $k_a$ is the association rate constant, $k_d$ is the dissociation rate constant, and $K_D$ is the equilibrium dissociation constant.
[c]$\chi^2$ values were derived from curve fitting using the first-order (1:1) Langmuir interaction model, which represent the average squared residual (i.e., the difference between the experimental data and the fitted curve).
[d]The kinetic constant $k_d$ was outside the limits of the instrument.

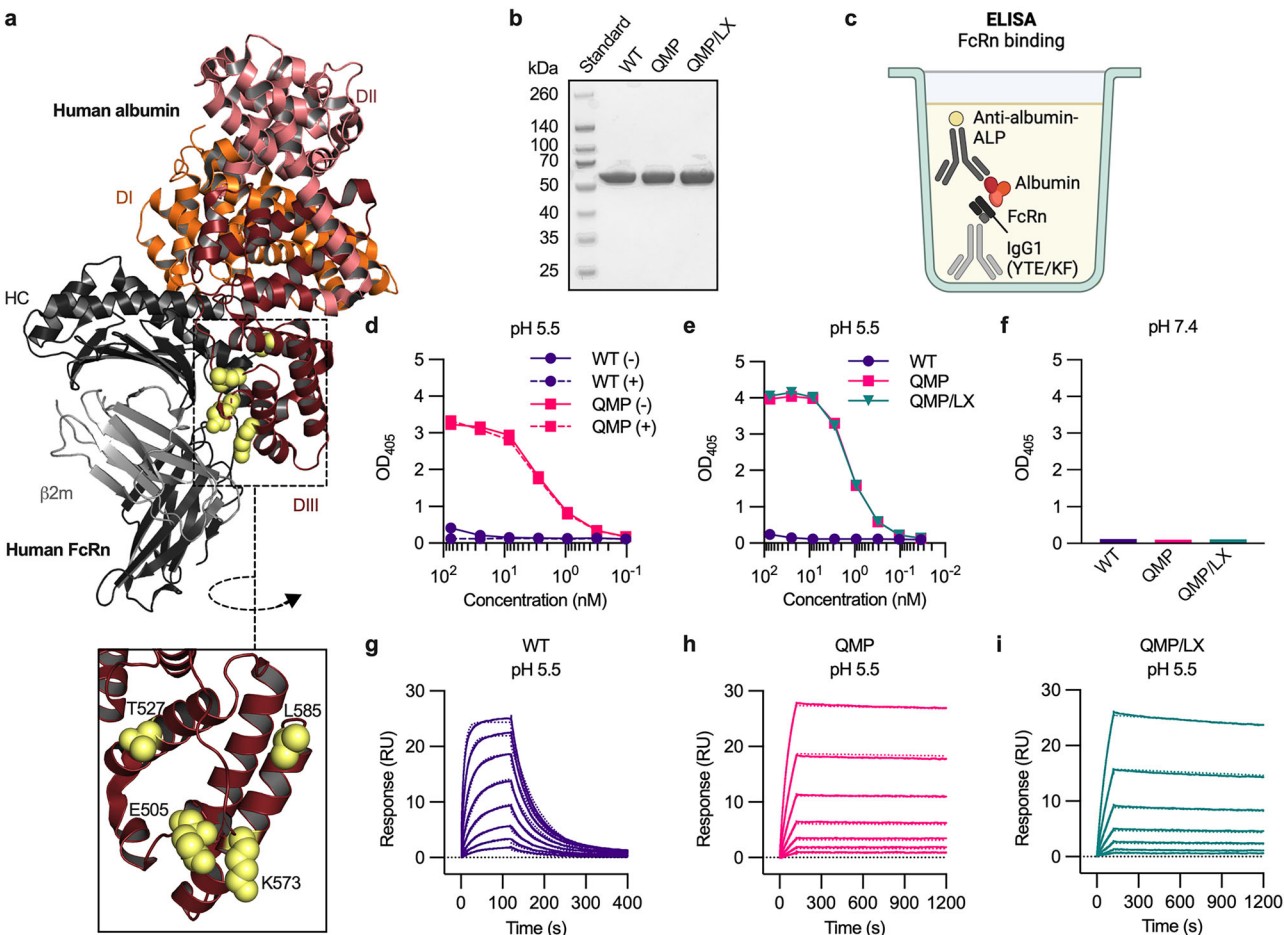

**Fig. 2 | QMP substitutions increase FcRn binding of albumin lacking L585.**
**a** Illustration of a solved co-crystal structure of WT human albumin in complex with human FcRn. DI, DII, and DIII of human albumin are shown in orange, salmon pink, and ruby, respectively. The truncated HC and β2m subunit of human FcRn are colored in black and gray, respectively. A close-up of the C-terminal end of DIII, where the three amino acid residues substituted in QMP (E505, T527, and K573), and L585, are shown as yellow spheres. The illustration was made using PyMOL with the solved crystal structure of human albumin in complex with human FcRn (PDB ID: 4N0F)[49]. **b** A non-reducing SDS-PAGE gel showing purified fractions of recombinant full-length WT albumin and QMP variants (66.5 kDa) produced in Expi293F cells. **c** Illustration of the ELISA setup used to measure FcRn–albumin binding. His-tagged human FcRn (black/gray) was captured on a coated human

IgG1 mutant variant (YTE/KF) (light gray), before human albumin (red) was added and detected using an ALP-conjugated anti-human albumin antibody (yellow-dark gray). The illustration was created with BioRender[100]. **d** Binding of WT albumin (purple) and QMP (pink), either not exposed (-) (solid lines) or exposed (+) (dotted lines) to CPA, to FcRn at pH 5.5. The values represent the mean from one representative experiment (*n* = 2). **e, f** ELISA results showing the pH-dependent FcRn binding of WT albumin (purple), QMP (pink), and QMP/LX (teal) at (e) pH 5.5 and (f) pH 7.4. The values represent the mean from one representative experiment (*n* = 2). **g–i** Representative SPR sensorgrams showing the binding of serial dilutions of **g** WT albumin (39–5000 nM), **h** QMP (1.6–50 nM), and **i** QMP/LX (1.6–50 nM) injected in duplicates over immobilized biotinylated human FcRn at pH 5.5 (solid lines). The data were fitted to the 1:1 Langmuir interaction model (dotted lines).

**Table 2 | Melting temperatures and elution pH for albumin variants and GLP-1 albumin fusions**

| Variant | $T_m$ (°C)[a] | Elution pH[b] |
|---|---|---|
| WT | 63.7 | 6.2 |
| QMP | 63.1 | – |
| QMP/LX | 64.2 | – |
| Tanzeum | 58.8 | 6.9 |
| GLP-1-WT | 57.4 | 6.7 |
| GLP-1-LX | 57.6 | 6.3 |
| GLP-1-QMP | 56.4 | 7.8 |
| GLP-1-QMP/LX | 56.8 | 7.7 |

[a]$T_m$ is the melting temperature determined at pH 7.4. The values represent the mean from one representative experiment (*n* = 2).
[b]pH values at the elution peak on the analytical FcRn affinity chromatography.

reduced binding to FcRn at pH 5.5 in ELISA (Fig. 4c, d). In contrast, GLP-1 QMP maintained its receptor binding irrespective of L585 cleavage (Fig. 4d). Similar results were obtained for two other human albumin fusion designs, where either recombinant coagulation factor IX (FIX)[9] or angiotensin-converting enzyme 2 (ACE2)[13] was genetically fused to the N-terminal end of human albumin (Supplementary Fig. 7). As for the GLP-1 albumin fusions, exposure to CPA resulted in removal of L585 and reduced FcRn binding of the WT albumin fusions, but not for the QMP-engineered counterparts (Supplementary Fig. 7, Supplementary Table 2).

The effect of L585 cleavage on FcRn binding was further investigated in ELISA with the recombinantly expressed full-length and truncated GLP-1 albumin fusions (Fig. 4c). At pH 5.5, GLP-1 WT showed slightly weaker binding to FcRn than Tanzeum, whereas GLP-1 LX demonstrated reduced receptor binding (Fig. 4e). This aligns with observations made when the GLP-1 albumin fusions were cleaved at the C-terminal end upon exposure to CPA. None of the albumin fusions bound the receptor at pH 7.4 (Fig. 4f). Notably, the QMP substitutions greatly enhanced FcRn binding of both GLP-1 QMP and GLP-1 QMP/LX at pH 5.5 (Fig. 4e). At pH 7.4, weak

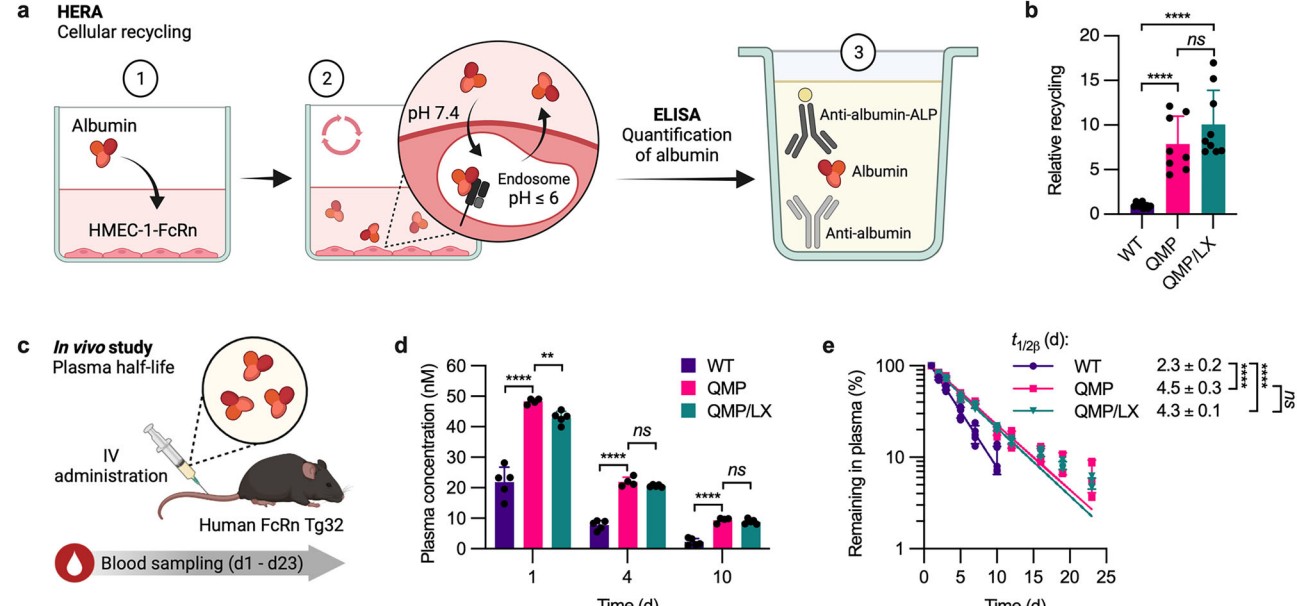

**Fig. 3 | Lack of L585 does not affect cellular recycling and plasma half-life of QMP-engineered albumin. a** Illustration of the HERA setup used to study FcRn-mediated cellular recycling. (1) Human albumin (red) was incubated with adherent HMEC-1-FcRn cells. (2) The amount of albumin recycled back to the culture medium was quantified by (3) ELISA, using an anti-human albumin antibody (light grey) and an ALP-conjugated anti-human albumin antibody (yellow-dark grey). **b** Cellular recycling of QMP (pink) and QMP/LX (teal), relative to WT albumin (purple). The values represent the mean ± SD from three independent experiments ($n = 3$). **c** Illustration of the in vivo experimental setup used to determine the plasma half-life of human albumin variants in hemizygous human FcRn Tg32 mice. Proteins were administered by IV injection into the lateral tail vein on day 0, and blood

samples were collected for up to 23 days. **d** Plasma concentrations of WT albumin (purple), QMP (pink), and QMP/LX (teal) on day 1, 4, and 10 post-administration. The values represent the mean ± SD from one representative experiment ($n = 5$ for WT albumin and QMP/LX; $n = 4$ for QMP). **e** Elimination curves of WT albumin (purple), QMP (pink), and QMP/LX (teal). The data are shown as the percentage of albumin remaining in plasma compared to day 1 and represent the mean ± SD from one representative experiment ($n = 5$ for WT albumin and QMP/LX; $n = 4$ for QMP). The plasma half-lives ($t_{1/2\beta}$) are shown as the mean ± SD. Unpaired two-tailed $t$-tests were used for statistical analysis, where $ns$ = not significant, $**p = 0.002$, and $****p < 0.0001$. Illustrations in **a** and **c** were created with BioRender[100].

binding of GLP-1 QMP was detected, whereas no binding was observed for GLP-1 QMP/LX (Fig. 4f).

Next, the binding kinetics and affinity at pH 5.5 were determined using SPR, as before, where the GLP-1 albumin fusions were injected over biotinylated human FcRn immobilized on an SA sensor chip (Fig. 4g). The results showed that Tanzeum bound with slightly faster association and dissociation rates than GLP-1 WT, resulting in a 1.5-fold difference in binding affinity to the receptor (Tanzeum: 76.1 nM; GLP-1 WT: 114.0 nM) (Fig. 4h, i, Table 1). For GLP-1 LX (514.9 nM), a 4.5-fold weaker binding affinity was measured compared to that of its full-length counterpart (Fig. 4j, Table 1), consistent with our previous report[60]. In comparison to the WT albumin fusions, GLP-1 QMP and GLP-1 QMP/LX demonstrated remarkably faster association and slower dissociation (Fig. 4k, l, Table 1). While the kinetic rate constants could not be accurately determined for GLP-1 QMP, GLP-1 QMP/LX (0.2 nM) showed an almost 600-fold higher binding affinity compared to GLP-1 WT (Fig. 4i, l, Table 1). Importantly, the fusion proteins that remained bound to the immobilized receptor at the end of the dissociation phase, rapidly dissociated upon injection of a pH 7.4 buffer during the regeneration phase (Supplementary Fig. 3d–h).

To evaluate receptor binding as a function of pH, and thus, mimic the pH gradient of the endosomal pathway, we performed analytical human FcRn affinity chromatography (Fig. 4m)[75]. Here, binding between the GLP-1 albumin fusions and column-bound human FcRn was initiated at pH 5.5, before dissociation was induced by gradually increasing the pH of the running buffer up to pH 8.8. Consistent with the binding affinities acquired by SPR, GLP-1 LX exhibited the weakest binding and dissociated from the column first at pH 6.3, followed by GLP-1 WT at pH 6.7 and Tanzeum at pH 6.9 (Fig. 4n, Table 2). Later dissociation was observed for GLP-1 QMP and

GLP-1 QMP/LX, which occurred at pH 7.8 and pH 7.7, respectively (Fig. 4n, Table 2). For comparison, unfused WT albumin dissociated at pH 6.2 (Fig. 4n, Table 2). Thus, the GLP-1 WT albiglutide-mimic has comparable FcRn binding characteristics as Tanzeum, while truncation of L585 from the C-terminal end of albumin decreases binding to the receptor. Notably, QMP engineering ensures favorable pH-dependent FcRn binding, also in the absence of the C-terminal leucine residue.

## QMP engineering extends the plasma half-life of the GLP-1 albumin fusion

Next, we examined whether QMP engineering of the GLP-1 albumin fusion would translate into extended plasma half-life, also after removal of L585. First, the FcRn-mediated cellular recycling properties of the full-length and truncated GLP-1 albumin fusions were evaluated by HERA with adherent HMEC-1-FcRn cells, as described (Fig. 5a). Following the uptake phase, 1.7-fold more of Tanzeum than GLP-1 WT was detected intracellularly, whereas the amount of GLP-1 LX was similar to that of GLP-1 WT (Supplementary Fig. 6b, d). Both GLP-1 QMP and GLP-1 QMP/LX were detected at about 3-fold higher concentrations than that of GLP-1 WT (Supplementary Fig. 6b, d). Furthermore, there was no significant difference in cellular recycling between Tanzeum and GLP-1 WT, while the effect of L585 truncation was apparent, with 2-fold less of GLP-1 LX being recycled compared to the WT counterpart (Fig. 5b, Supplementary Fig. 6f). In contrast, GLP-1 QMP/LX was recycled equally well as GLP-1 QMP, about 4.5-fold more efficiently than GLP-1 WT (Fig. 5b, Supplementary Fig. 6f).

To evaluate the PK properties, the GLP-1 albumin fusions were administered to human FcRn Tg32 mice by IV injections, and blood samples were collected for up to 12 days (Fig. 5c). Following the initial 1-day distribution phase, about 22–26% of the administered Tanzeum and GLP-1 WT were detected in plasma (Supplementary Fig. 8a). The plasma

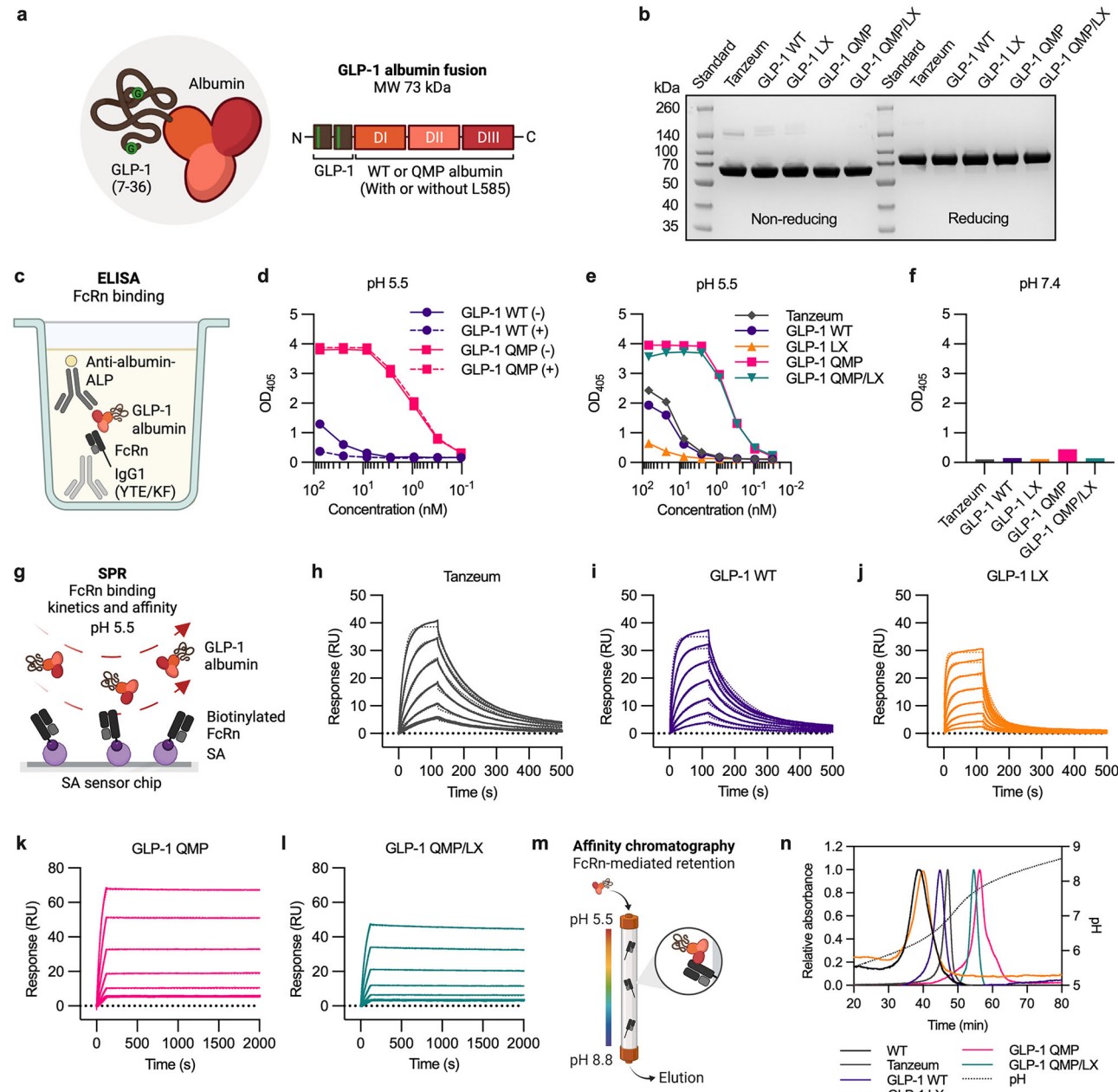

**Fig. 4 | QMP-engineered GLP-1 albumin fusions show strong and pH-dependent binding to human FcRn. a** Illustration of the GLP-1 albumin fusion (73 kDa), where two sequential copies of the GLP-1 (residues 7–36) peptide hormone (brown) containing an alanine-to-glycine substitution at position 2, shown as green circles, are genetically fused to the N-terminal DI of human albumin (orange). DI, DII, and DIII of albumin are shown in orange, salmon pink, and ruby, respectively. The designed GLP-1 albumin fusions contain either full-length or truncated versions of WT albumin or QMP. **b** An SDS-PAGE gel showing purified fractions of Tanzeum and the GLP-1 albumin fusions produced in Expi293F cells, under non-reducing and reducing conditions. **c** Illustration of the ELISA setup used to examine FcRn–albumin binding. His-tagged human FcRn (black/gray) was captured on a coated human IgG1 mutant variant (YTE/KF) (light gray), before the GLP-1 albumin fusions (brown-red) were added and detected using an ALP-conjugated anti-human albumin antibody (yellow/dark gray). **d** Binding of GLP-1 WT (purple) and GLP-1 QMP (pink), not exposed (-) (solid lines) or exposed (+) (dotted lines) to CPA, to FcRn at pH 5.5. The values represent the mean from one representative experiment (n = 2). **e, f** FcRn binding of Tanzeum (gray), GLP-1 WT (purple), GLP-

1 LX (orange), GLP-1 QMP (pink), and GLP-1 QMP/LX (teal) at **e** pH 5.5 and **f** pH 7.4. The values represent the mean from one representative experiment (n = 2). **g** Illustration of the SPR setup used to measure FcRn binding kinetics and affinity at pH 5.5, where biotinylated human FcRn (purple-black/grey) was immobilized on an SA sensor chip, followed by injection of GLP-1 albumin fusions (brown-red). **h–l** Representative SPR sensorgrams showing the binding of serial dilutions of **h** Tanzeum (19.5–625 nM), **i** GLP-1 WT (19.5–1250 nM), **j** GLP-1 LX (39–5000 nM), **k** GLP-1 QMP (1.6–50 nM), and **l** GLP-1 QMP/LX (1.6–50 nM) injected in duplicates over immobilized FcRn at pH 5.5 (solid lines). The data were fitted to the 1:1 Langmuir interaction model (dotted lines). **m** Illustration of the analytical FcRn affinity chromatography setup used to evaluate the interaction between GLP-1 albumin fusions and column-incorporated human FcRn across a pH-gradient of 5.5 to 8.8. **n** Chromatograms showing the elution of WT albumin (black), Tanzeum (gray), GLP-1 WT (purple), GLP-1 LX (orange), GLP-1 QMP (pink), and GLP-1 QMP/LX (teal). The pH as a function of time is shown (dotted line). Illustrations in **a**, **c**, and **m** were created with BioRender[100].

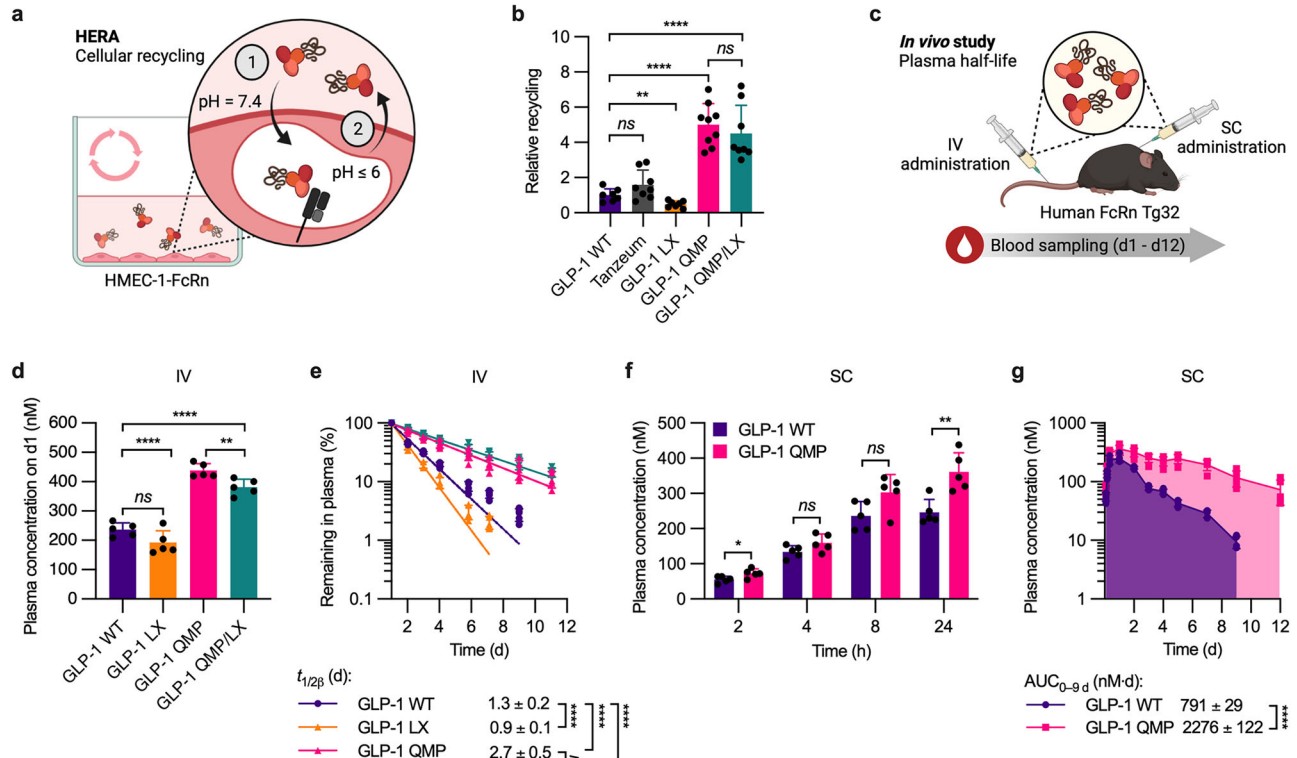

**Fig. 5 | QMP-engineered albumin improves the PK properties of fused GLP-1.**
**a** Illustration of the HERA with adherent HMEC-1-FcRn cells used to study FcRn-mediated cellular recycling of GLP-1 albumin fusions. **b** Cellular recycling of Tanzeum (grey), GLP-1 LX (orange), GLP-1 QMP (pink), and GLP-1 QMP/LX (teal), relative to GLP-1 WT (purple). The values represent the mean ± SD from three independent experiments ($n = 3$). Unpaired two-tailed $t$-tests were used for statistical analysis, where $ns$ = not significant, $**p = 0.006$, and $****p < 0.0001$.
**c** Illustration of the in vivo experimental setup used to evaluate the PK properties of GLP-1 albumin fusions in homozygous human FcRn Tg32 mice. The fusion proteins were administered by IV injection into the lateral tail vein, or by SC injection into the scruff of the neck. **d** Plasma concentrations of GLP-1 WT (purple), GLP-1 LX (orange), GLP-1 QMP (pink), and GLP-1 QMP/LX (teal) on day 1 following IV administration. The values represent the mean ± SD from one representative experiment ($n = 5$). Unpaired two-tailed $t$-tests were used for statistical analysis,

where $ns$ = not significant, $**p = 0.0077$, and $****p < 0.0001$. **e** Elimination curves of GLP-1 WT (purple), GLP-1 LX (orange), GLP-1 QMP (pink), and GLP-1 QMP/LX (teal) following IV administration. The data are shown as the percentage of GLP-1 albumin fusion remaining in plasma compared to day 1 and represent the mean ± SD from one representative experiment ($n = 5$). The plasma half-lives ($t_{1/2\beta}$) are shown as the mean ± SD. Unpaired two-tailed $t$-tests were used for statistical analysis, where $ns$ = not significant and $****p < 0.0001$. **f, g** Plasma concentrations of GLP-1 WT (purple) and GLP-1 QMP (pink) (f) at 2, 4, 8, and 24 h and **g** up to 12 days following SC administration. The values represent the mean ± SD from one experiment ($n = 5$). The AUC values are shown as the mean ± SD. Unpaired two-tailed $t$-tests were used for statistical analysis, where $ns$ = not significant, $*p = 0.0405$, $**p = 0.0046$, and $****p < 0.0001$. Illustrations in **a** and **c** were created with BioRender[100].

concentrations of the GLP-1 WT albiglutide-mimic and Tanzeum were comparable at the following time points, resulting in similar PK properties (Supplementary Fig. 8b, Supplementary Table 4). For GLP-1 LX, about 22% of the administered dose was detected in plasma 1 day post-administration (Fig. 5d). Following the distribution phase, GLP-1 LX was eliminated more rapidly than GLP-1 WT, yielding a shorter plasma half-life of 0.9 days compared to 1.3 days (Fig. 5e, Supplementary Table 4). In agreement, a 1.4-fold faster CL and a 1.6-fold shorter MRT were calculated for GLP-1 LX compared to GLP-1 WT with gPKPDSim (Supplementary Table 4). In comparison to GLP-1 WT, GLP-1 QMP and GLP-1 QMP/LX showed about 2-fold higher plasma concentrations, with about 48% and 42% of the administered doses detected in the blood after the distribution phase, respectively (Fig. 5d, Supplementary Table 4). The plasma half-lives of both GLP-1 QMP (2.7 days) and GLP-1 QMP/LX (3.3 days) were about 2-fold longer than that of GLP-1 WT (1.3 days) (Fig. 5e). Moreover, about 3-fold slower CL and 3-fold longer MRT were calculated for the QMP-engineered fusions compared to GLP-1 WT (Supplementary Table 4). Thus, truncation of the C-terminal L585 residue of the GLP-1 fusion containing WT albumin reduces its cellular recycling in vitro and plasma half-life in vivo, which may be compensated for by QMP engineering of albumin, irrespective of the presence of L585.

## QMP engineering improves the PK properties of the GLP-1 albumin fusion following subcutaneous administration

Subcutaneous (SC) injection is the most common route of administration for GLP-1 RAs, including Tanzeum[24,25]. To study systemic distribution of the GLP-1 albumin fusions following SC administration, we measured the presence of the proteins in the blood for up to 12 days post-injection into the scruff of the neck of human FcRn Tg32 mice (Fig. 5c). Comparable plasma concentrations of the GLP-1 WT and GLP-1 QMP fusions were detected at 2, 4, and 8 h post-injection (Fig. 5f, g). After 24 h, the plasma concentration of GLP-1 QMP peaked, being 1.5-fold higher than that of GLP-1 WT, corresponding to 40% and 27% of the administered doses, respectively (Fig. 5f, g, Supplementary Table 4). The following 8 days, the plasma concentrations of GLP-1 QMP were consistently higher than those of GLP-1 WT, resulting in a 3.1-fold greater AUC value (Fig. 5g, Supplementary Table 4). Similar to when administered by IV, GLP-1 QMP showed a 2.1-fold extended plasma half-life compared to GLP-1 WT (Supplementary Table 4). In agreement, a 3.5-fold reduced CL and a 2.4-fold longer MRT were calculated for GLP-1 QMP (Supplementary Table 4). Thus, QMP engineering improves the PK properties of the GLP-1 albumin fusion both when administered by SC and IV injections.

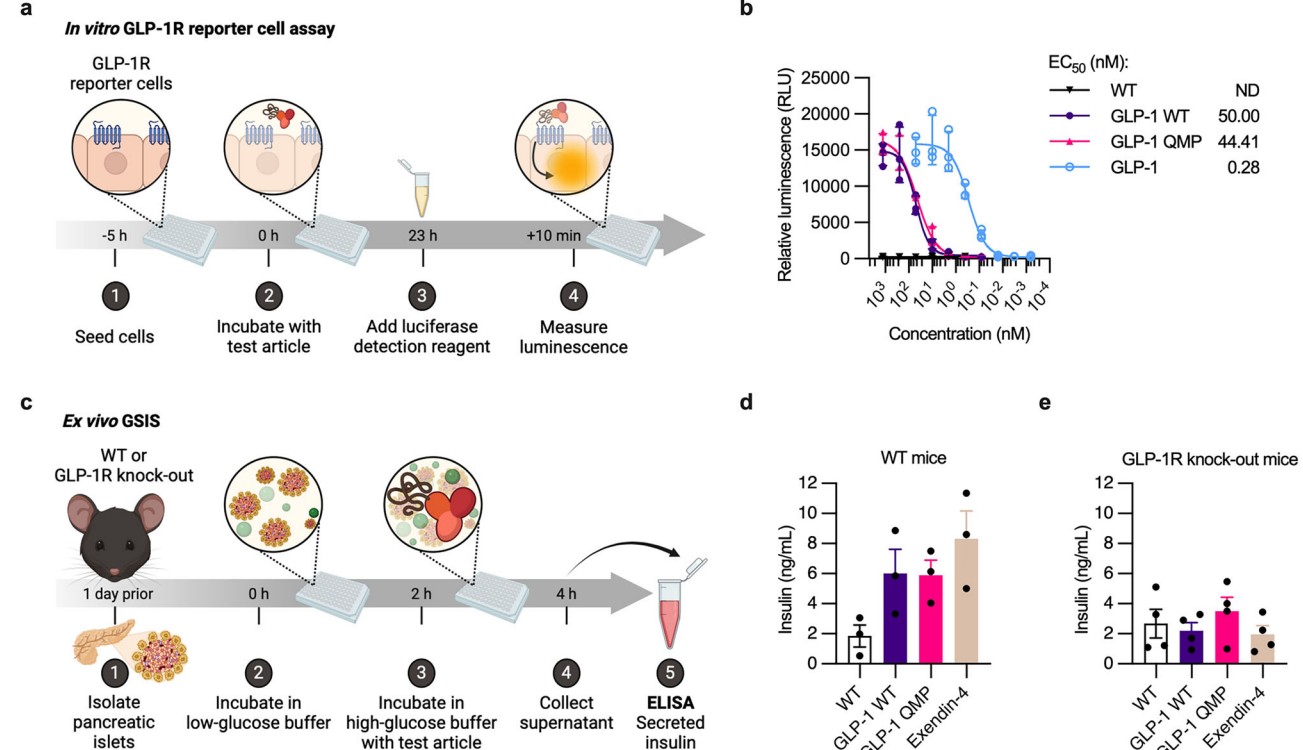

**Fig. 6 | GLP-1 albumin fusions activate the GLP-1R signaling pathway and stimulate glucose-dependent insulin secretion in pancreatic islet cells. a** Illustration of the in vitro GLP-1R reporter cell assay. (1) GLP-1R reporter cells were seeded in a 96-well plate and incubated for 5 h, (2) before the test articles were added. (3) After 23 h, the luciferase detection reagent was added, and (4) the luminescence was measured after 10 min. **b** Relative luminescence measured after incubation with WT albumin (black), GLP-1 WT (purple), GLP-1 QMP (pink), or unfused GLP-1 (blue). The values represent the mean ± SD from one representative experiment ($n = 3$). The mean EC$_{50}$ values are shown. ND = not determined. **c** Illustration of the ex vivo GSIS assay performed with primary pancreatic islets isolated from WT or GLP-1R knock-out mice. (1) Pancreatic islet cells were isolated from the mice and

incubated overnight. (2) The islets were incubated in a low-glucose buffer for 2 h, before the supernatants were collected and (3) replaced with a high-glucose buffer containing test articles. (4) After 2 h, the supernatants were collected. (5) ELISA was performed to quantify the secreted insulin concentrations. **d, e** Concentrations of insulin secreted by islets isolated from **d** WT mice or **e** GLP-1R knock-out mice after incubation with WT albumin (black), GLP-1 WT (purple), GLP-1 QMP (pink), or Exendin-4 (beige). The values represent the mean ± SD from one experiment ($n = 3$ for WT mice, $n = 4$ for GLP-1R knock-out mice), calculated by subtracting the insulin concentrations measured after incubation in the low-glucose buffer from those measured after incubation in the high-glucose buffer (GSIS values). Illustrations in **a** and **c** were created with BioRender[100].

## GLP-1 albumin fusions stimulate insulin secretion through a GLP-1R-dependent mechanism

To test the ability of the GLP-1 albumin fusions to activate the GLP-1R in vitro, we used a human GLP-1R reporter cell assay kit, where serial dilutions of unfused GLP-1, GLP-1 WT, or GLP-1 QMP were incubated with the cells overnight (Fig. 6a). On the following day, a luciferase detection reagent was added, before luminescence was measured. GLP-1 WT and GLP-1 QMP gave rise to overlapping dose-dependent response curves with EC$_{50}$ values of 50.00 nM and 44.41 nM, respectively (Fig. 6b). In comparison, a lower EC$_{50}$ value of 0.28 nM was calculated for unfused GLP-1 (Fig. 6b). Thus, the GLP-1 WT albiglutide-mimic and GLP-1 QMP fusions engaged the human GLP-1R expressed by the cells, although at a lower potency than unfused GLP-1, which is in line with previous reports on albiglutide/Tanzeum and other GLP-1 fusion proteins[10,76,77].

Next, we performed an ex vivo cell-based glucose-stimulated insulin secretion (GSIS) assay using primary pancreatic islet cells isolated from either WT mice or GLP-1R knock-out mice (Fig. 6c)[78]. When incubated with either GLP-1 WT, GLP-1 QMP, or Exendin-4, the pancreatic islet cells isolated from the WT mice secreted similar amounts of insulin, 3 to 4.5-fold more than when incubated with unfused WT albumin (Fig. 6d). In comparison, the pancreatic islets isolated from the GLP-1R knock-out mice secreted 2 to 4-fold less insulin after incubation with GLP-1 WT, GLP-1 QMP, or Exendin-4, similar amounts to that measured for unfused WT albumin (Fig. 6e). Thus, the designed GLP-1 albumin fusions stimulate glucose-dependent insulin secretion in a GLP-1R-dependent manner.

Importantly, the QMP substitutions did not affect the ability of GLP-1 to activate the GLP-1R.

## QMP engineering prolongs the ability of the GLP-1 albumin fusion to restrict glycemic excursion in vivo

To investigate whether the GLP-1 albumin fusions would stimulate insulin secretion in vivo, we performed a glucose tolerance test (GTT) in human FcRn Tg32 mice (Fig. 7a). To mimic an overnight fast in humans, the mice were fasted for 4 hours in the morning[79], after which initial blood glucose levels were measured. Subsequently, the mice were given either Tanzeum or GLP-1 WT by intraperitoneal (IP) injection, while a control group received an equimolar amount of unfused WT albumin. After 1 hour, the blood glucose levels were measured immediately before administering a high dose of glucose by IP injection. Blood glucose levels were monitored at 10, 20, 30, 60, and 90 min post-injection. The glycemic excursion in the groups receiving Tanzeum and GLP-1 WT was comparable, and much lower than in the control group (Fig. 7b). Moreover, mice given either fusion consistently demonstrated an increase in the plasma concentrations of insulin following the glucose challenge (Fig. 7c). Thus, the GLP-1 WT albiglutide-mimic demonstrated in vivo functionality and glucose-lowering properties on par with those of Tanzeum.

To determine whether the improved PK properties of the QMP-engineered fusions would translate into enhanced efficacy, we repeated the GTT in human FcRn Tg32 mice. In this experiment, the mice received the GLP-1 WT albiglutide-mimic or GLP-1 QMP 6 h before injection of a high-

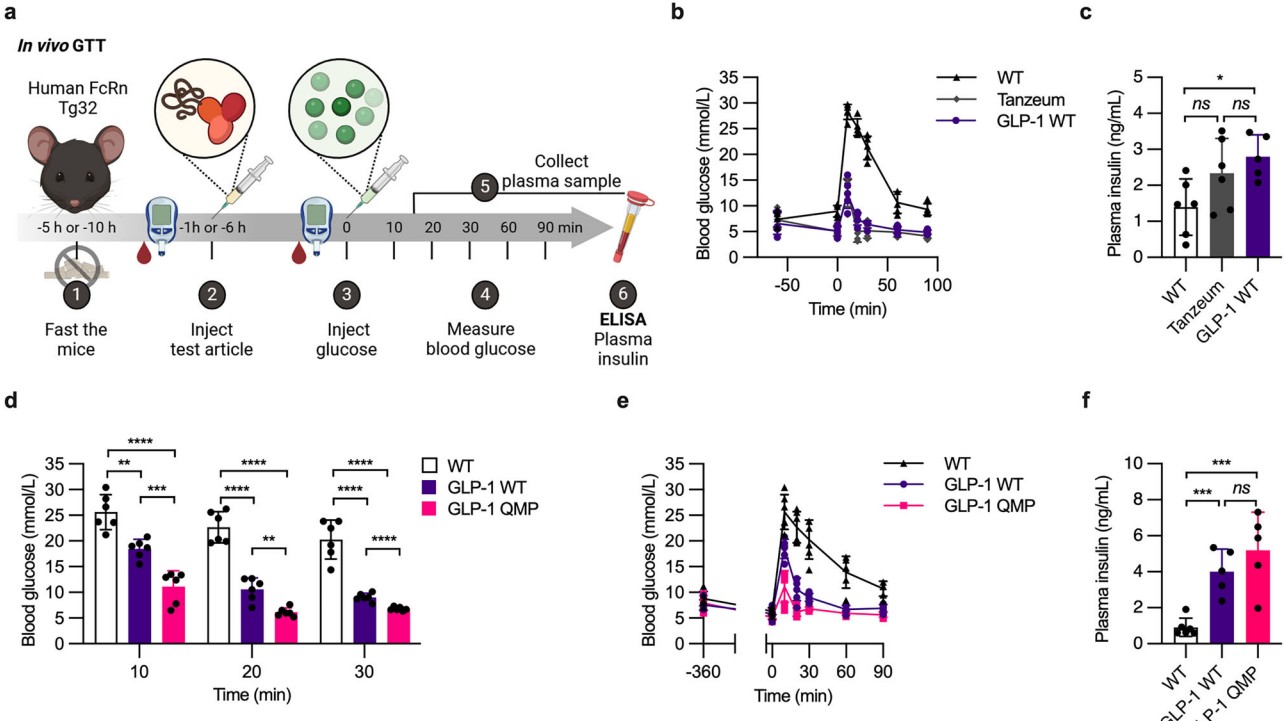

**Fig. 7 | QMP-engineered GLP-1 albumin improves glycemic control in mice.**
**a** Illustration of the GTT performed in human FcRn Tg32 mice. (1) After a 4-h fasting period, (2) the blood glucose levels were measured using a glucometer, before the test articles were administered to the mice by IP injections. (3) After 1 or 6 h, the blood glucose levels were measured, immediately before the mice received a high dose of glucose by IP injections. (4) Blood glucose levels were measured 10, 20, 30, 60, and 90 min post-glucose injection, and (5) blood samples were collected between 10 and 20 min after glucose injection. (6) ELISA was performed to quantify the plasma insulin concentrations. The illustration was created with BioRender[100]. **b** Blood glucose concentrations following administration of a high dose of glucose to human FcRn Tg32 mice receiving WT albumin (black), Tanzeum (gray), or GLP-1 WT (purple) 1 h prior to the glucose load. The values represent the mean ± SD from one

representative experiment (*n* = 5–6). **c** Plasma insulin concentrations between the 10 and 20-min time points following the glucose load. The values represent the mean ± SD from one representative experiment (*n* = 5–6). **d, e** Blood glucose concentrations after **d** 10, 20, and 30 min, and **e** up to 90 min, following administration of a high dose of glucose, in human FcRn Tg32 mice that received WT albumin (black), GLP-1 WT (purple), or GLP-1 QMP (pink) 6 h prior to the glucose load. The values represent the mean ± SD from one representative experiment (*n* = 6).
**f** Plasma insulin concentrations between the 10 and 20-min time points following the glucose load. The values represent the mean ± SD from one representative experiment (*n* = 6). Unpaired two-tailed *t*-tests were used for statistical analysis, where *ns* = not significant, **p* = 0.0101, ***p* = 0.0011, ****p* ≤ 0.0009, and *****p* < 0.0001.

dose glucose (Fig. 7a). The group receiving GLP-1 QMP displayed a notably reduced glycemic spike compared to those that received GLP-1 WT or unfused WT albumin, with an associated increase in plasma insulin concentrations (Fig. 7d–f). Specifically, 10 min after glucose injection, the blood glucose levels in the group that received GLP-1 QMP were almost 2-fold lower than in the group receiving GLP-1 WT (Fig. 7d, e). Furthermore, the glucose levels in the former group returned to baseline within 20 min, while it took over 30 min in the latter group (Fig. 7d, e). Thus, QMP engineering prolongs the glucose-lowering effect of the GLP-1 albumin fusion.

## Discussion
Genetic fusion of protein-based drugs to the N-terminal or C-terminal end of albumin, or both, offers a feasible and attractive approach to enhance the PK properties and therapeutic efficacy of short-lived biologics[6,7]. Despite the susceptibility of C-terminal cleavage, using albumin for half-life extension of biologics remains appealing for several reasons. Firstly, albumin is a non-glycosylated, single-chain polypeptide[1], which facilitates expression and manufacturing of monovalent albumin fusions. Secondly, unlike the Fc fragment of IgG antibodies, albumin does not engage effector molecules of the immune system, thereby minimizing the risk of potential adverse effects[1].

While several albumin fusions are currently in preclinical development or clinical trials, two have so far reached the market[7]. Two years after albiglutide (Tanzeum), approval of a FIX albumin fusion, albutrepenonacog alfa (Idelvion), for treatment of hemophilia B, followed[80]. Both albumin

fusions extend the plasma half-lives of GLP-1 and FIX by manyfold[30–32,81,82]. However, the average plasma half-lives of albiglutide (5 days)[30–32] and albutrepenonacog alfa (3–4 days)[81,82] in humans are considerably shorter than that of endogenous albumin (20 days)[5]. Consistent with the human PK data, we observed that the plasma half-life of GLP-1 WT in transgenic mice expressing human FcRn was 1.8-fold shorter than that of unfused WT albumin. Engagement of its target receptor, the GLP-1R, is likely one factor contributing to the faster clearance rate for the GLP-1 albumin fusion. Moreover, given the high plasma concentrations of endogenous albumin (40 mg/mL)[1], the competitive pressure on the albumin binding site of FcRn is substantial[11]. Inevitably, even minor alterations in FcRn binding properties of a protein fusion can therefore affect the ability of the receptor to rescue it from intracellular lysosomal degradation. Unexpectedly, we found that GLP-1 WT bound human FcRn 5-fold more strongly than unfused WT albumin at acidic pH, due to both faster association and slower dissociation. In this case, the binding was measured towards a recombinant truncated and soluble form of the receptor. When we evaluated FcRn-mediated recycling in a cell-based assay, where the receptor is embedded in the membrane, the WT fusion was recycled 1.8-fold less efficiently than unfused WT albumin, which is in line with its shorter plasma half-life. In this regard, it is interesting that IgG antibodies that show similar FcRn binding properties can have variable plasma half-lives due to differences in biophysical properties, such as pI and net charge, which affect cellular handling[83–87]. However, as the GLP-1 albumin fusion and unfused albumin have the same theoretical pI, and only small differences in pH-dependent net charge, this

probably does not explain the discrepancy between their FcRn binding properties and cellular handling or plasma half-life.

In humans, L585 cleavage by CPA reduces the plasma half-life of albumin from 3 weeks to 3.5 days due to loss of FcRn binding affinity[60]. In the two approved albumin fusions, GLP-1 and FIX are fused to the N-terminal end of albumin due to functional requirements of the biologics[66,67,88–91]. Such fusion design leaves the C-terminal end of albumin susceptible for enzymatic cleavage. As anticipated, we found that the L585 residue on the C-terminal end of the GLP-1 albumin fusion was removed upon exposure to CPA, which decreased FcRn binding. This was also the case for two designed FIX and ACE2 albumin fusion proteins. Consequently, the plasma half-life of the GLP-1 albumin fusion was reduced from 1.3 days to 0.9 days in the human FcRn Tg32 mice. Thus, removal of the C-terminal L585 reduces the ability of the albumin fusions to engage human FcRn, and thereby compromises their rescue from intracellular lysosomal degradation.

Although not applicable for GLP-1, fusion to the C-terminal end of albumin could be an attractive approach to avoid cleavage by CPA. However, this strategy is only attractive if the therapeutic agent itself is not a substrate for CPA, or if the case, that its functional activity is not negatively affected by C-terminal cleavage. Moreover, it is important that FcRn engagement is not hampered upon fusion to the C-terminal DIII of albumin, which contains the main binding site for the receptor[52,58]. We previously found that fusion of a peptide or a single-chain variable fragment to the C-terminal end of albumin reduced FcRn binding up to 2-fold, and to a greater extent than when fused to the N-terminal end[12]. However, this may vary depending on the nature of the fused biologic. Therefore, the orientation should be considered on a case-to-case basis upon designing new albumin fusions.

To ensure optimal FcRn binding, cellular transport, and PK properties of albumin-based biologics, we attempted to substitute the C-terminal L585 of albumin with an amino acid residue that would not be recognized by CPA. Interestingly, L585 could only be replaced with an amino acid having similar physiochemical properties, including isoleucine, methionine, and valine, without negatively affecting FcRn binding[64]. We previously demonstrated that the L585 residue of albumin promotes a structural destabilization in three regions of the C-terminal DIII domain, and thus, induces an intra-domain flexibility required for efficient FcRn binding[60]. This study further supports that this local destabilization is dependent on the nature of a non-polar branched or medium-long aliphatic amino acid side chain. However, all such amino acid residues are recognized by CPA[64]. Accordingly, substituting the C-terminal L585 residue of albumin with an amino acid resistant to the enzymatic cleavage by CPA proved challenging without compromising receptor engagement.

Further, we explored whether QMP engineering in DIII of albumin could restore its receptor binding upon L585 cleavage. As demonstrated, introduction of the QMP substitutions resulted in a favorable receptor binding, efficient FcRn-mediated cellular recycling, and consequently, extended plasma half-life both in the unfused albumin and GLP-1 albumin fusion formats. Strikingly, these favorable features conferred by QMP were independent of the presence of a C-terminal L585. Thus, QMP engineering of albumin evidently provides a competitive advantage in receptor binding throughout the endosomal pathway, thereby favoring rescue from intracellular degradation in vivo. Importantly, the QMP-engineered GLP-1 albumin fusion did not only display improved PK properties, but also demonstrated a prolonged therapeutic effect in healthy human FcRn Tg32 mice challenged with a high dose of glucose. Specifically, it was shown to potently stimulate insulin release in response to glucose, and to restrict glycemic excursion, beyond that of the GLP-1 albumin fusion composed of WT albumin, likely due to a higher concentration in plasma. Nevertheless, to further assess the impact of QMP engineering of the GLP-1 albumin fusion on glycemic control and treatment of T2DM, additional studies are needed in relevant disease models.

One alternative approach to direct fusion to albumin, in which one would not have the risk of C-terminal end cleavage limiting the therapeutic potential, is by fusing or conjugating GLP-1 or other therapeutic agents to an albumin-binding molecule[7]. This strategy has been exploited in the long-acting GLP-1RA semaglutide (Novo Nordisk), for instance, which was approved for treatment of T2DM in 2017, and for treatment of obesity in 2021, and has since then become the most prescribed GLP-1RA[92]. In semaglutide, DPP-4 resistant GLP-1 is conjugated to a fatty acid chain of 18 carbon atoms, a natural ligand of albumin[93]. The drug therefore non-covalently associates with endogenous albumin upon administration, which extends the plasma half-life of the peptide from a few hours to 7 days, enabling once-weekly injections[93,94]. However, it also means that the strategy is limited by the plasma half-life of endogenous WT albumin.

In the current study, we show that QMP engineering for enhanced pH-dependent FcRn engagement extends the plasma half-life of albumin fusion proteins far beyond that achieved by using WT albumin. Importantly, we demonstrate that the QMP substitutions in DIII of albumin compensate for the reduction in FcRn binding and receptor-mediated cellular recycling caused by enzymatic cleavage of the C-terminal end. This has implications for how to design human albumin-fused biologics, as QMP engineering may ensure a long plasma half-life, increased target exposure, and enhanced efficacy, despite potential C-terminal cleavage post-administration.

## Materials and methods
### Structural inspection
Coordinates of the solved crystal structures of human albumin (PDB ID: 6WUW)[95] and truncated human FcRn in complex with human albumin (PDB ID: 4N0F)[52] were retrieved and inspected using PyMOL (version 2.5) (Schrödinger, Inc.).

### Vector construction
Previously described expression vectors (pcDNA3.1) encoding human albumin variants (WT, LX, and QMP) were utilized[11,48,60]. cDNA fragments encoding DIII of albumin with either of the naturally occurring amino acids, except for cysteine, in position 585, or QMP/LX, were synthesized and sub-cloned using the restriction enzyme sites *BamHI* and *XhoI* (GenScript).

Expression vectors (pFUSE2ss) encoding GLP-1 fused to the N-terminal end of albumin were constructed by sub-cloning cDNA fragments encoding two tandem GLP-1 peptide sequences (residues 7–36), both with the A2G substitution, in-frame of cDNA encoding human albumin variants (WT, LX, QMP, or QMP/LX) using the restriction enzyme sites *EcoRI* and *NheI* (GenScript). The expression vectors for the FIX albumin fusion variants were kindly provided by Alessio Branchini (University of Ferrara, Italy)[9].

### Production and purification of albumin variants and albumin fusion proteins
Albumin, either unfused or genetically fused to GLP-1, FIX[9], or ACE2[13], were produced using the Expi293 Expression System (Gibco), according to the manufacturer. Briefly, suspension Expi293F cells (cat. no. A14527, Gibco) were cultured in Expi293F Expression Medium (Gibco) in baffled flasks on an orbital shaker platform (Thermo Fisher Scientific) set to 125 RPM at 37 °C and 8% $CO_2$. The cells ($3 \times 10^6$ cells/mL) were transfected with 1 μg of plasmid DNA per mL of cell culture using ExpiFectamine 293 Reagent (Gibco). ExpiFectamine Transfection Enhancers I and II (Gibco) were added 18–22 h later. Six days post-transfection, the cell culture supernatant was harvested by centrifugation at ($290$–$400$) $\times g$ for 25 min at 4 °C and filtered using a 0.2-μm Vacuum Filtration System (VWR).

The secreted proteins were purified from the harvested supernatants using a CaptureSelect Human Albumin Affinity Matrix (Thermo Fisher Scientific) packed in a 5-mL OPUS Chromatography Column (Repligen)[48]. The column was pre-equilibrated with 10 column volumes (CV) of 1× Dulbecco's Phosphate Buffered Saline (DPBS) (pH 7.4) (Sigma-Aldrich)/ 0.05% (v/v) sodium azide (Sigma-Aldrich), before the supernatant was loaded onto the column at a flow rate of (0.5–2) mL/min. The column was then washed with 20 CV of DPBS/0.05% (v/v) sodium azide, followed by

elution with 9 CV of 2 M MgCl$_2$ (Sigma-Aldrich)/20 mM Tris-HCl (Sigma-Aldrich) buffer (pH 7.4) at a flow rate of 2 mL/min. The eluted proteins were buffer-exchanged to DPBS and concentrated using Amicon Ultra-15 Centrifugation Filter Units (Millipore) with a 30 kDa cutoff by centrifugation at (290–400) × $g$ at 4 °C. Subsequently, SEC was conducted on an ÄKTA avant 25 instrument (Cytiva) coupled to a Superdex 200 Increase 10/300 GL Column (Cytiva). Isolated monomeric protein fractions were concentrated using Amicon Ultra-4 30k Centrifugation Filter Units (Millipore), and the final protein concentration was determined using a DS-11+ Spectrophotometer (DeNovix).

### Production and purification of recombinant human FcRn

Recombinant soluble His-tagged human FcRn was produced using a Baculovirus expression vector system[96]. Briefly, semi-adherent High Five cells (cat. no. B85502, Invitrogen) cultured in flat culture flasks were grown until confluent in Express Five SFM Medium (Gibco) supplemented with 200 mM L-glutamine (Sigma-Aldrich) and 1× Antibiotic-Antimycotic solution (Sigma-Aldrich) at 27 °C. Next, the cells ($1 \times 10^6$ cells/mL) were transferred to Erlenmeyer flasks and transfected with a viral stock encoding the His-tagged soluble human FcRn kindly provided by Dr. Sally Ward (Cancer Sciences Unit, Centre for Cancer Immunology, University of Southampton, Southampton, United Kingdom), and further incubated for 72 h at 23–24 °C and 160 RPM.

The cell culture supernatant was harvested by centrifugation and filtered using a 0.2 µm filter. Secreted receptor was purified from the supernatant using a 1-mL HisTrap HP Column (Cytiva) pre-charged with Ni$^{2+}$ ions. The column was pre-equilibrated with 10 CV of DPBS (Sigma-Aldrich)/0.05% (v/v) sodium azide (Sigma-Aldrich), before the supernatant adjusted to pH 7.2 was loaded onto the column. The column was then washed with 10 CV of 25 mM imidazole (Sigma-Aldrich) in DPBS (pH 7.2), followed by elution with 10 CV of 250 mM imidazole in DPBS (pH 7.3). The flow rate was set to 5 mL/min in all steps. To obtain monomeric fractions, SEC was performed on a Superdex 200 Increase 10/300 GL Column (Cytiva), as described above[97].

### SDS-PAGE

Non-reducing and reducing SDS-PAGE analyses were performed using a Bolt 12% Bis-Tris Plus Gel (Invitrogen). For non-reducing SDS-PAGE, 2 µg of protein was mixed with 3 µL of 4× Bolt LDS Sample Buffer (Invitrogen) and Milli-Q water to a total volume of 12 µL. For reducing SDS-PAGE, 2 µg of protein was mixed with 3 µL of 4× Bolt LDS Sample Buffer (Invitrogen), 1.2 µL of 10× Bolt Sample Reducing Agent (Invitrogen), and Milli-Q water to a total volume of 12 µL, and the samples were incubated for 10 min at 95 °C. The protein samples (12 µL) and Spectra Multicolor Broad Range Protein Ladder (7.5 µL) with 10–260 kDa range (Thermo Fisher Scientific) were loaded onto the gel, and electrophoresis was run at 200 V for 22 min in 1× Bolt MES SDS Running Buffer (Invitrogen). The gel was rinsed three times with Milli-Q water, stained for 30 min with Bio-Safe Coomassie Brilliant Blue (Bio-Rad Laboratories), and destained overnight in Milli-Q water.

### SPR

SPR was performed using a Biacore T200 instrument (Cytiva). Recombinant biotinylated human FcRn (Immunitrack) (1 µg/mL) was immobilized (~100 RU) on a Sensor Chip SA (Cytiva), following the manufacturer's protocol. The receptor was diluted and injected in 1× HBS-P+ Buffer (0.01 M HEPES, 0.15 M NaCl, 0.05% (v/v) Surfactant P20; pH 7.4) (Cytiva). A single concentration (5 µM) or 2-fold serial dilutions of unfused albumin variants (78–5000 nM) or GLP-1 albumin fusions (1.6–5000 nM) were injected over the immobilized FcRn at a flow rate of 40 µL/min at 25 °C. PBS (177 mM phosphate, 85 mM NaCl, 0.05% (v/v) Tween 20) at pH 5.5 was used as dilution and running buffer, while PBS (195 mM phosphate, 85 mM NaCl, 0.05% (v/v) Tween 20) at pH 7.4 was used as regeneration buffer. All sensorgrams were zero-adjusted, and the reference cell response was subtracted. Kinetic and affinity constants were derived using the Langmuir 1:1

ligand binding model provided by the Biacore T200 Evaluation Software (version 3.0; Cytiva).

### CPA digestion of albumin

To initiate in vitro enzymatic cleavage by CPA, unfused albumin (WT or QMP) or albumin fused to GLP-1, FIX[9], or ACE2[13] (15 µM), in DPBS (pH 7.4) (Sigma-Aldrich), were incubated with 30 µg/mL of human pancreatic CPA (Elastin Products Company, Inc.) for 5 h at 37 °C on a shaker platform set to 450 RPM. The proteins were stored at −80 °C until analysis.

### LC-MS/MS

LC-MS/MS was used to analyze the protein variants either exposed to or not exposed to CPA[60]. Briefly, 3 µg of each protein sample was mixed with 15 µL of 50 mM ammonium bicarbonate (pH 7.8) (Thermo Fisher Scientific), followed by protein reduction with 2 µL of 100 mM dithiothreitol (Thermo Fisher Scientific) for 30 min and protein alkylation with 5 µL of 2-iodoacetamide (Thermo Fisher Scientific) for an additional 30 min at room temperature in the dark. For protein digestion, 0.5 µg of trypsin (Promega) was added to the samples, followed by an overnight incubation at 37 °C. Further, the digested peptides were desalted using the sample preparation protocol from Evosep Biosystems.

LC-MS/MS was performed using an Evosep LC (Evosep Biosystems) connected to a Q Exactive HF (Thermo Fisher Scientific). All analyses were operated in data-dependent mode, where the most intense peptides were automatically selected for fragmentation by high-energy collision-induced dissociation. Protein identification and quantification were determined using MaxQuant Software (version 2.0.3.0) (Max Planck Institute of Biochemistry), by referencing the sequences of WT albumin, LX, QMP, and QMP-LX. The following parameters were used: carbamidomethylation as a fixed modification, protein N-acetylation and methionine oxidation as variable modifications, a first search error window of 20 ppm, and a main search error of 4.5 ppm. Trypsin with two miscleavages allowed was used. To quantify peptide level, the intensity of the peptide lacking L585 was compared to that of the peptide containing L585 within the same sample. Correlation of possible differences in ionization efficiency of each peptide was not conducted.

### nanoDSF

nanoDSF was performed using a Prometheus NT.48 instrument (NanoTemper Technologies GmbH). Unfused albumin variants or GLP-1 albumin fusions (1 mg/mL), diluted in DPBS (pH 7.4) (Sigma-Aldrich) were loaded into Prometheus Grade Standard Capillaries (NanoTemper Technologies GmbH). The excitation power was set to 80%, and the temperature was set to increase from 20 °C to 95 °C with a temperature ramp rate of 1 °C per min. $T_m$ was determined by the built-in PR.ThermControl Software (version 2.3.1) (NanoTemper Technologies GmbH).

### Prediction of pI and net charge

The EMBOSS iep software (EMBOSS) (https://www.bioinformatics.nl/cgi-bin/emboss/iep?_pref_hide_optional=0) was used to calculate the theoretical pI and net charge at different pH values of unfused albumin variants and GLP-1 albumin fusions.

### ELISA for FcRn binding analysis

96-well EIA/RIA Clear Flat Bottom Polystyrene Microplates (Corning) were coated overnight at 4 °C with 8 µg/mL (100 µL/well) of a recombinant Fc-engineered human IgG1 mutant (YTE/KF)[98], with specificity for 4-hydroxy-3-iodo-5-nitrophenylactic acid, diluted in DPBS (pH 7.4) (Sigma-Aldrich). The wells were blocked with DPBS (pH 7.4)/4% (w/v) skimmed milk powder (Sigma-Aldrich) (PBS/M) (250 µL/well) for 1 h at room temperature and washed three times with PBS/0.05% (v/v) Tween 20 (PBS/T) at either pH 5.5 or pH 7.4. Next, 10 µg/mL (100 µL/well) of His-tagged human FcRn, diluted in PBS/T/M (pH 5.5 or pH 7.4), was added for 1 h at room temperature, followed by washing as before. Serial dilutions of unfused albumin variants (0.0343–75.2 nM) or GLP-1 albumin fusions

(0.0313–68.5 nM) (100 µL/well), diluted in PBS/T/M (pH 5.5 or pH 7.4), were added for 1 h at room temperature. After washing, 125 ng/mL (100 µL/well) of ALP-conjugated anti-human albumin polyclonal antibody from goat (cat. no. A80-229AP, Bethyl Laboratories, Inc.), diluted in PBS/T/M (pH 5.5 or pH 7.4), was added for 1 h at room temperature, followed by a washing step and detection with 1 mg/mL (100 µL/well) of *p*-Nitrophenyl Phosphate Substrate (Sigma-Aldrich) dissolved in diethanolamine buffer (pH 9.8) (Sigma-Aldrich). Absorbance was measured at 405 nm using a Sunrise Absorbance Microplate Reader (Tecan Group Ltd.).

### Analytical FcRn affinity chromatography
Analytical FcRn affinity chromatography was performed using a human FcRn Affinity Column[75] (Roche), coupled to an ÄKTA avant 25 instrument (Cytiva). Unfused albumin variants or GLP-1 albumin fusions (2 mg/mL) were loaded onto the column in a pH gradient from pH 5.5 to pH 8.8 by injecting a gradual mixture of 20 mM MES/HCl, 140 mM NaCl (Sigma-Aldrich) (pH 5.5) and 20 mM Tris/HCl, 140 mM NaCl (Sigma-Aldrich) (pH 8.8) at a flow rate of 0.5 mL/min for 110 min at room temperature. Absorbance at 280 nm and pH were monitored.

### ELISA for albumin quantification
Albumin concentration in HERA and plasma samples was quantified by ELISA. For HERA samples, 96-well EIA/RIA Clear Flat Bottom Polystyrene Microplates (Corning) were coated with 8 µg/mL (100 µL/well) of anti-human albumin polyclonal antibody from goat (cat. no. A1151, Sigma-Aldrich), diluted in DPBS (pH 7.4) (Sigma-Aldrich), and incubated overnight at 4 °C. For plasma samples, 1 µg/mL of an anti-human albumin monoclonal antibody from mouse (15C7) (cat. no. MA1-90420, Invitrogen; cat. no. Ab10241, Abcam) was used. On the subsequent day, the wells were blocked (200 µL/well) with PBS/M (pH 7.4) for 1 h at room temperature and washed (250 µL/well) four times with PBS/T (pH 7.4). Protein samples or albumin standards (100 µL/well), diluted in PBS/T/M (pH 7.4), were added for 1 h at room temperature, before washing as before. Next, 125 ng/mL (100 µL/well) of ALP-conjugated anti-human albumin polyclonal antibody from goat (cat. no. A80-229AP, Bethyl Laboratories, Inc.) in PBS/T/M was added for 1 h at room temperature, followed by wash and detection with 1 mg/mL (100 µL/well) of *p*-Nitrophenyl Phosphate Substrate (Sigma-Aldrich) dissolved in diethanolamine buffer (pH 9.8) (Sigma-Aldrich). Absorbance was measured at 405 nm using a Sunrise Absorbance Microplate Reader (Tecan Group Ltd.).

### HERA
Adherent HMEC-1-FcRn cells[69] provided by Dr. Wayne I. Lencer (Boston Children's Hospital, Harvard Medical School and Harvard Digestive Diseases center, United States) were cultured in MCDB 131 medium (Gibco) supplemented with 10% (v/v) heat-inactivated fetal bovine serum (FBS) (Gibco), 2 mM L-glutamine (Gibco), 25 µg/mL streptomycin (Gibco), 25 U/mL penicillin (Gibco), 10 ng/mL recombinant mouse epidermal growth factor (Gibco), and 1 µg/mL hydrocortisone (Sigma-Aldrich), at 37 °C in a humidified atmosphere of 5% CO₂. To maintain a stable expression of human FcRn, 250 µg/mL Geneticine selective antibiotic (G418 Sulfate) (Gibco) and 5 µg/mL Blasticidin S HCl (Gibco) were added to the culture medium.

HERA[48] was performed by seeding HMEC-1-FcRn cells ($1.5 \times 10^5$ cells/well) to a Costar 48-well TC-Treated Multiple Well Plate (Corning) in complete culture medium (250 µL/well) for approximately 20–24 h until confluent. The cells were washed three times with pre-warmed Hank's Balanced Salt Solution (HBSS) (Gibco) (200 µL/well) and starved for 1 h in HBSS (250 µL/well). Next, 800 nM of protein samples (125 µL/well), diluted in HBSS, was added to the cells. After 3 h of incubation, the cells were washed five times with cold HBSS, and immediately stored at −20 °C until analysis. A parallel plate treated identically was incubated for an additional 3 h in fresh assay medium (complete culture medium, without FBS, G418 Sulfate, and Blasticidin S HCl, supplemented with 1× Eagle's MEM Non-Essential Amino Acids Solution; Gibco) (220 µL/well). The assay media

were collected and stored at −20 °C. The plate containing the frozen cells were lyzed with Pierce RIPA Lysis and Extraction Buffer (Thermo Fischer Scientific) supplemented with 1× cOmplete Protease Inhibitor Cocktail (Roche) (220 µL/well) for 10 min on ice on a tilting board. To remove cell debris, the plates were centrifuged for 10 min at 3500 RPM and 4 °C, before the supernatants were collected. The cell lysates (uptake samples) and collected assay media (recycling samples) were analyzed by ELISA.

### In vitro cell-based human GLP-1R reporter assay
An in vitro cell-based assay was performed using a Human GLP-1R Reporter Assay Kit (cat. no. IB33001, INDIGO Biosciences, Inc.) according to the manufacturer. Briefly, the GLP-1R reporter cells were seeded into white-bottom 96-well assay plates in cell recovery medium (200 µL/well) and incubated at 37 °C in a humidified atmosphere of 5% CO₂ for 5 h. Wells containing medium only were included to quantify the plate-specific fluorescence background signal. The medium was then removed, and serial dilutions of the protein samples (200 µL/well) were added to the cells (unfused WT albumin, GLP-1 WT, and GLP-1 QMP: 0.08–6250 nM; or GLP-1(7–36) included in the kit: 0.00062–50 nM). After 23 h of incubation, the supernatants were removed, and a luciferase detection reagent (100 µL/well) was added. The plate was incubated at room temperature for 5–10 min, before luminescence was measured using a Victor³ 1420 Multilabel Counter (Wallac 1420 Workstation, version 3.00 revision 5; PerkinElmer) set at 5 s shaking and 0.5 s reading.

### Ex vivo pancreatic islet GSIS assay
Pancreatic islets were isolated from WT or GLP-1R knockout C57BL/6 mice[99] using collagenase perfusion[78]. Briefly, the islets were incubated in RPMI 1640 Medium (Gibco) at 37 °C overnight, before they were washed in Krebs-Ringer HEPES (KRH) solution with 2.8 mM glucose (low-glucose buffer). Five islets per group were incubated at 37 °C for 2 h in the low-glucose buffer, before the supernatants were collected and stored until analysis. The islets were further incubated in 50 µL of KRH solution with 16.7 mM glucose (high-glucose buffer), supplemented with 2 µM of unfused WT albumin, GLP-1 WT, GLP-1 QMP, or Exendin-4 (Byetta) (positive control). The islets were incubated at 37 °C for 2 h, before the supernatants were collected and stored until analysis. Insulin secreted from the islets into the medium was quantified using an ultra-sensitive mouse insulin ELISA kit (cat. no. 90080, Crystal Chem, Inc.) according to the manufacturer. GSIS values were calculated by subtracting the insulin concentrations measured after incubation in the low-glucose buffer from those measured in the high-glucose buffer.

### Animal experiments
Animal experiments were performed using hemizygous or homozygous human FcRn Tg32 mice (B6.Cg-*Fcgrt*$^{tm1Dcr}$ Tg(FCGRT)32Dcr/DcrJ, The Jackson Laboratory). The in vivo half-life study in hemizygous human FcRn Tg32 mice was performed at The Jackson Laboratory (JAX Services, Bar Harbor, ME, USA), which was approved by the Animal Care and Use Committee at The Jackson Laboratory. All other animal experiments were carried out at the Section of Comparative Medicine, Oslo University Hospital Rikshospitalet (Oslo, Norway) upon approval by the Norwegian Food Safety Authority (FOTS ID 23998). We have complied with all relevant ethical regulations for animal use. All animals (up to 5 mice per cage) were housed under controlled conditions in ventilated cages, including a 12-h light/dark cycle, temperatures of $22 \pm 4$ °C or $21 \pm 2$ °C, and relative humidity ranges of $50 \pm 15\%$ or 30–70% at The Jackson Laboratory and Oslo University Hospital Rikshospitalet, respectively. The bedding and nesting material were changed on a regular basis, and the mice were given *ad libitum* access to water and food.

### In vivo half-life studies
In hemizygous human FcRn Tg32 mice, groups of male mice (7–8 weeks, 21.5–30.5 g, 4–5 mice/cage/group) were given IV injection, into the lateral

tail-vein, of 1 mg/kg of unfused WT albumin and QMP albumin variants, diluted in DPBS (pH 7.4) (Sigma-Aldrich). Blood (25 μL) was collected from the retro-orbital sinus for up to 23 days. Blood samples were mixed with 1 μL of 1% K3 EDTA. Plasma was isolated by centrifugation at 17,000 × g for 5 min at 4 °C and diluted 1:10 in 50% (v/v) glycerol in PBS (pH 7.4) (Sigma-Aldrich) for storage at −20 °C until analysis.

In homozygous human FcRn Tg32 mice, groups of female and male mice (8–16 weeks, 21.5–30.0 g, 5 mice/2–3 cages/group) were given either IV or SC injection, into the scruff of the neck, of 4 mg/kg of GLP-1 albumin fusions, diluted in DPBS (pH 7.4) (Sigma-Aldrich), respectively. Blood (25 μL) was collected from the saphenous vein using minicaps Na-hep Micro Capillary Pipettes (Hirschmann Laborgeräte) for up to 12 days. Plasma was isolated by centrifugation at 17,000 × g for 10 min at 4 °C and stored at −20 °C until analysis.

Protein concentrations in plasma samples were analyzed by ELISA and presented as the percentage of protein remaining at specific time points post-administration compared to the protein concentration on day 1 (set as 100%). Linear regression analysis (line fitting) was performed using GraphPad Prism Software (version 9.5.1) (GraphPad Software, Inc.). The $\beta$-phase half-life was calculated using the following formula:

$$t_{1/2\beta} = \log \frac{0.5}{A_e/A_0} \times t,$$

where $t_{1/2\beta}$ is the β-phase half-life, $A_e$ is the protein concentration at a specified post-injection time point, $A_0$ is the protein concentration on day 1, and $t$ is the elapsed time.

The concentrations of the albumin variants in plasma from all animal studies were applied to the gPKPDSim application in MATLAB (version R2020b; MathWorks)[74] for prediction of PK parameters.

### In vivo GTT

GTT experiments were performed in homozygous human FcRn Tg32 mice. Groups of male mice (12–13 weeks, 22.5–28.5 g, 5–6 mice/2–3 cages/group) were fasted for 4 h prior to IP injection of unfused WT albumin (0.9 mg/kg) or GLP-1 albumin fusions (1 mg/kg). After 1 or 6 h, the mice received 1.5 g/kg of glucose solution (Gibco) by IP injection. Blood was collected from the saphenous vein, and blood glucose levels were measured using an ACCU-CHEK Guide Glucose Monitor and Testers (Roche) before each injection and at 10, 20, 30, 60, and 90 min after the glucose injection. Plasma from blood samples (50 μL) collected between the 10 and 20-min time point was isolated by centrifugation at 17,000 × g for 10 min at 4 °C and stored at −20 °C until analysis.

### ELISA for insulin quantification

Insulin concentration in the plasma samples collected during the GTTs was determined using a Mouse Ultrasensitive Insulin ELISA Kit (cat. no. 80-INSMSU-E01, Alpco), according to the manufacturer's instruction. Briefly, plasma samples, as well as standard and control samples, were added (5 μL/well) to a 96-well microplate pre-coated with a monoclonal antibody specific for insulin. Working Strength Conjugate was then added (75 μL/well) and incubated for 2 h at room temperature on a microplate shaker set to 750 RPM. The wells were washed (350 μL/well) six times with Working Strength Wash Buffer and incubated for 30 min with TMB substrate (100 μL/well), before Stop Solution (100 μL/well) was added. Absorbance was measured at 450 nm using a Sunrise Absorbance Microplate Reader (Tecan Group Ltd.).

### Statistics and reproducibility

All data were processed, analyzed, and visualized using GraphPad Prism 9 Software (version 9.5.1) (GraphPad Software, Inc.). HERA was performed three times with three wells per test article each time. GLP-1R reporter assay was performed twice with triplicates for each test article concentration. Ex vivo pancreatic islet GSIS assay was performed once with three to four pancreatic islets isolated from different mice. PK evaluation of unfused albumin variants and GLP-1 albumin fusions following IV injection was performed twice with four or five mice per test article. PK evaluation of GLP-1 albumin fusions following SC injection was performed once with five mice per test article. GTT experiments were performed twice with five or six mice per test article. Potential outliers were identified using the ROUT method with coefficient Q set to 1% or 2% in the GraphPad Prism 9 Software. Statistical significance was determined using unpaired student's $t$-tests, with a 95% confidence level and $p < 0.05$ defined as statistically significant. Illustrations were created with BioRender.com.

### Reporting summary

Further information on research design is available in the Nature Portfolio Reporting Summary linked to this article.

### Data availability

Source data underlying graphs presented in the main figures can be found in Supplementary Data 1 and are available from the corresponding author on reasonable request. Uncropped SDS-PAGE gels can be found in Supplementary Information (Supplementary Fig. 9). The MS-based proteomics data have been deposited to the ProteomeXchange Consortium (http://proteomecentral.proteomexchange.org) via the PRIDE partner repository with the dataset identifier PXD063561.

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

## Acknowledgements
The work was partially supported by the Research Council of Norway through its Centres of Excellence scheme, project number 332727, and grant 274993 (J.T.A., J.N., and S.B.), grant 287927 (J.T.A., K.H.A., and F.R.J.), and grant 285136 (J.T.A.). The work was also supported by the South-Eastern Norway Regional Health Authority; grant 2024046 (K.H.A., D.B., and J.T.A.), and grants 2018052 and 2019084 (J.T.A.). MS-based proteomic analyses were performed by the Proteomics Core Facility, Department of Immunology, University of Oslo/Oslo University Hospital, which is supported by the Core Facilities program of the South-Eastern Norway Regional Health Authority. This core facility is also a member of the National Network of Advanced Proteomics Infrastructure (NAPI), which is funded by the Research Council of Norway INFRASTRUKTUR-program (project number: 295910).

## Author contributions
J.N., K.H.A., S.B., I.S. and J.T.A. designed research; J.N., K.H.A., S.B., F.R.J., T.U.G., D.B., M.S. and S.S. performed research; T.S. provided instrumentation; J.N., K.H.A., S.B., T.U.G., M.S., S.S., T.S. and J.T.A. analyzed data; J.N., K.H.A., S.B. and J.T.A. wrote the paper. All authors have read and approved the final version of the paper.

## Competing interests
I.S. and J.T.A. are co-inventors of the patents entitled "Albumin Variants and Uses Thereof" (e.g., EP3063171B1, US10208102, and US10781245). The remaining authors declare no competing interests.
