## [Transparent Peer Review file · Communications Biology]

Enhanced plasma half-life and efficacy of engineered human albumin-fused GLP-1 despite enzymatic cleavage of its C-terminal end

Corresponding Author: Professor Jan Terje Andersen

Version 0:

Reviewer comments:

Reviewer #1

(Remarks to the Author)

The submitted manuscript reported an approach to fuse GLP-1 with human albumin of its C-terminal end to enhance plasma half-life and efficacy. This is a very interesting strategy to secure a long plasma half-life and enhanced efficacy of fused biologics in vivo. At the current stage, the authors shall address the comments below.

1. It is unclear the effect of 3 residues QMP on their enhanced efficiency. Will the single residue of QMP have a similar outcome?
2. In the assessment of QMP enhanced cellular uptake, it would be better to use endocytosis inhibitor further to study the QMP variant cellular uptake.

Reviewer #2

(Remarks to the Author)

Andersen and co-workers describe an approach to use an "engineered" human serum albumin to connect to make it last longer in vivo. Albumin is degraded by carboxypeptidase A (CPA) by removal of L585 that seems to be necessary for binding to FcRn (neonatal Fc receptor). The lifecycle and homeostasis of albumin is dictated by binding to FcRn (after pinocytosis) in intracellular endosomal compartments. If albumin is bound at pH < 6.5 to FcRn, it is trafficked back to the plasma membrane rather than being targeted for degradation in lysosomes.

The main point of the paper is this. The above narrative that they have explored in previous papers can be exploited to create an albumin for attachment to therapeutic peptides and proteins and enhance their half-lives if the attachment does not interfere with the therapeutic peptide's function.

First the authors show through SPR experiments that L585 cannot be substituted away easily by mutagenesis as it also affects binding to FcRn. This was shown by creating 18 variants of albumin replacing L585. Only Ile, Met, and Val substitutions (of L585) resulted in variants that had good affinity for FcRn. But the dilemma is that these are also removed by CPA and diminish the binding to FcRn.

Instead they resort to a previously described three-mutation "albumin QMP" (QMP for three of the residues replaced) leaving L585 intact. The salient feature that is being exploited is that somehow even if L585 is lopped off of the QMP-albumin, it retains its ability to bind FcRn at pHs < 6.5 and therefore should retain the recycling back to the plasma membrane and be released. QMP resulted from an earlier investigation from the same group (ref. 62) so that aspect is not novel.

Major:

The SPR studies are rigorously done and the paper is well written. The in vivo work and affinity data are convincing. So this reviewer is left with the following major concerns about novelty given ref. 62 (Sci. Transl. Med. 2020, 12, eabb0580) from the same group: the "QMP-albumin" has been already described. Addition of GLP-1 to QMP and similar results was certainly one would have expected. While the construct with GLP-1 attached follows the albiglutide example, pharmacology of GLP-1

at the GLP-1R is entirely missing. The authors must show that this construct that has already gone extensive in vivo lifetime evaluation and also shown to be recycled back to the plasma membrane and released is viable at the GLP-1R at potencies and efficacies rivaling the native peptide.

Minor points:

- (1) The investigators keep calling methionine a branched amino acid in the manuscript. The methionine side chain has no branching.
- (2) Page 4, lines 104-106: this sentence needs to be rewritten. The N-terminus of GLP-1 is indeed necessary for activity at the GLP-1R but the way this sentence is written, it seems that the N-terminus of albumin is attached to GLP-1 -- not sure how this makes sense.
- (3) Page 6, Lines 130-131: "CPA cleaves L585, but not the following glycine residue (G584)". Protease cleave in between residues. They don't cleave any individual "residues". The authors should state "that CPA catalyzes hydrolysis of the amide bond between L585 and ..."

Overall, this is a good paper and deserves publication with concerns about novelty and a very major concern experimentally about showing GLP-1 activity.

Reviewer #3

(Remarks to the Author)

It this article authors have presented progress on their work where they had identified the C-terminal amino acid of albumin important for its tight binding to FcRn and QMP mutation as beneficial in terms of increasing FcRn binding. The authors show in this paper that QMP can overcome the deletion of c-terminal leucine, and substitution of leucin with other amino acids that have branched aliphatic chain does not affect FcRn binding, They also show that fusion of QMP albumin with modified GLP-1 peptide retains the higher FcRn binding despite c-terminal leucine deletion. Lastly, PK and PD of GLP-1+albumin fused protein is shown to further support the conclusion. In general, the paper is well written, and only minor revisions are needed. Below are a few comments for the authors to address.

-While the authors have briefly mentioned that GLP-1 fused albumin construct had shorter half-life than just albumin, there is not mechanistic explanation provided for it. It will be good to show maybe FcRn binding data or some other properties of the molecules that leads to this difference in pharmacokinetic behavior.

-Please discuss what will happen if the proteins such as GLP-1 are fused to the C-terminal? Would they be cleaved in the presence of CPA, or will create steric hindrance for CPA and prevent c-terminal deletion.

-The conclusion that QMP provides better subcutaneous bioavailability seems incorrect based on figure 5g and supplementary table 3. After IV administration the ratio of QMP and WT albumin AUC is ~3.8, and that for QMP and WT fused proteins is ~2.7. After SC administration the ratio for QMP and WT fused protein AUCs is ~3.1. Thus, it does not seem bioavailability is changing much, and the apparent difference in AUC mainly stems from different PK. While higher binding to FcRn at pH 6 has been shown to provide better subcutaneous bioavailability of antibody, the data in this case does not seem convincing enough.

-Discussion is just repetition of results and can benefit from broader perspective of authors on implications of this work and its comparison with findings from other groups on similar subject matter.

Version 1:

Reviewer comments:

Reviewer #1

(Remarks to the Author)

The authors have addressed my previous concern and provided additional support.

Reviewer #2

(Remarks to the Author)

Unfortunately for this reviewer, the major points have not been addressed

- (1) The example of albiglutide vs the QMP-construct is not shown on long term receptor activation.
- (2) This paper is not only about function but mostly about longevity. That is still missing unfortunately in this reviewer's opinion.
- (3) The potency loss of the fused GLP1-albumin construct is 178 or 157-fold (0.28 nM vs 50 or 44 nM) -- that essentially means less than 0.03% of the activity. Is that relevant? This is why the albiglutide (~100 fold less potent) experiment and comparison is relevant.
- (4) Now there are new issues: Why would one expect enhanced insulin secretion with a fused albumin? "RESULTS" in response to reviewer 2.

The minor points seem to have been adequately addressed.

This reviewer is still unconvinced on the major points lifetime and recycling (major point of the paper) are addressed.

Reviewer #3

(Remarks to the Author)

Authors have addressed all the comments satisfactorily and manuscript has been improved notably. The reviewer has no further suggestions.

Version 2:

Reviewer comments:

Reviewer #2

(Remarks to the Author)

The authors have satisfactorily addressed the concerns of this reviewer

January 6, 2025

We thank the reviewers for their helpful and constructive comments. All modifications in the manuscript are highlighted in yellow. We have also included continuous line numbering in the manuscript and refer to page and line numbers in the point-by-point reply below.

Reviewer #1 (Remarks to the Author):

The submitted manuscript reported an approach to fuse GLP-1 with human albumin of its C-terminal end to enhance plasma half-life and efficacy. This is a very interesting strategy to secure a long plasma half-life and enhanced efficacy of fused biologics in vivo. At the current stage, the authors shall address the comments below.

1. It is unclear the effect of 3 residues QMP on their enhanced efficiency. Will the single residue of QMP have a similar outcome?

We thank the reviewer for the question. Indeed, the QMP-engineered GLP-1 albumin fusion protein demonstrated a prolonged therapeutic effect, as compared to the GLP-1 WT fusion, in human FcRn transgenic mice. This is likely a result of an extended plasma half-life, and thus, a higher plasma concentration of the QMP-engineered fusion protein at the time the glucose-tolerance test was initiated, rather than due to enhanced engagement of the GLP-1 receptor.

This is further supported by new data from two cell-based assays that we have included in the revised version of the manuscript, which show that both GLP-1 WT and GLP-1 QMP engage the GLP-1 receptor (GLP-1R) equally well, and promote glucose-dependent insulin secretion from mouse pancreatic islet cells in a receptor-dependent manner.

We have previously demonstrated that the single amino acid substitution, K573P, improves pH-dependent FcRn binding by 12-fold, which in turn, extends the plasma half-life of albumin by 1.5-fold in human FcRn Tg32 mice and cynomolgus monkeys (Andersen et al., JBC, 2014). Moreover, we have shown that combining K573P with the two amino acid substitutions, E505Q and T527M, further enhances FcRn-mediated recycling, extending the plasma half-life of albumin by 1.7-fold beyond than that of the K573P substitution alone (Bern et al., Sci. Transl. Med., 2020). In addition, we have previously shown that the QMP variant extends the plasma half-life of complex fusion partners, including coagulation factors VII and IX, and a decoy receptor, by up to 3.5-fold longer than WT albumin in human FcRn Tg32 mice (Lombardi et al., Br. J. Haem., 2021; Benjakul et al., PNAS Nexus, 2023).

Thus, to achieve enhanced therapeutic efficacy, the above data support that the GLP-1 albumin fusion protein would benefit more from introducing all three QMP substitutions, rather than introducing only one of the amino acid substitutions.

The data from the two cell-based assay have been added to the Results, pages 13 and 14, lines 359–379 (highlighted in yellow), and to a separate figure (Figure 6). The material and methods have been updated in accordance with the results, pages 27 and 28, lines 735–760 (highlighted in yellow).

2. In the assessment of QMP enhanced cellular uptake, it would be better to use endocytosis inhibitor further to study the QMP variant cellular uptake.

We thank the reviewer for the comment. In regard to cellular uptake, previous studies, including live-imaging on HMEC-1 overexpressing FcRn, as well as on primary endothelial cells, strongly support that IgG and albumin are taken up by non-specific fluid-phase pinocytosis (references 36-45 in the manuscript). As the interactions between FcRn and its ligands are pH dependent, the receptor does not play a significant role in the uptake of IgG and albumin into the endothelial cells. Binding does not occur in the neutral pH of the blood at the cell surface, but is first initiated after uptake, in the mildly acidic environment of endosomes, where FcRn is mainly located. Following engagement of FcRn, the ligands are recycled back to the cell surface, where exposure to the neutral pH of the blood again triggers release. Thus, pH-dependent binding is crucial for efficient FcRn-mediated recycling of IgG and albumin. Moreover, FcRn is mainly localized intracellularly within endosomal compartments, and is only transiently present on the cell surface.

Importantly, the QMP amino acid substitutions increase the binding of albumin to FcRn at acidic pH, without disrupting the pH-dependency of the interaction, and thus, should not affect uptake or release at the cell surface. In agreement, the results from HERA showed that the QMP-engineered albumin and GLP-1 albumin fusions were recycled 5 to 10-fold more efficiently than the WT counterparts.

To provide more insight into the cellular handling of QMP-engineered variants, we have now included additional results from the HERA assay. The new data show the amounts of protein taken up by the cells, in addition to the amounts that were recycled back out of the cells. Briefly, confluent adherent HMEC-1-FcRn cells were starved for 1 hour, before equimolar amounts of the unfused albumin variants or the GLP-1 albumin fusions were added and incubated for 3 hours to allow for cellular uptake. The cells were then lysed, and the amounts of albumin inside the cells were quantified by ELISA. The results showed 4-5-fold more efficient uptake for the QMP engineered albumin and albumin fusions compared to the corresponding WT proteins. It is important to keep in mind that this is a measure of the amount present inside the cells at one time point, while the cellular processes that occur are dynamic. During the 3-hour incubation step, albumin is taken up via fluid-phase pinocytosis (independent of FcRn), but also recycled via FcRn back to the cell surface, or alternatively, degraded if not captured by the receptor. Hence, the amount shown in the uptake graphs is the sum of these events, where increased FcRn-mediated recycling and rescue from lysosomal degradation contribute to the higher amounts of QMP-containing proteins than the WT counterparts inside the cells, rather than directly representing enhanced cellular uptake.

We appreciate the focus on the cellular uptake of QMP, and agree that the new data further strengthen our manuscript. *The HERA data on the cellular uptake have been added to the Results, pages 8 and 9, lines 217–220, and pages 11 and 12, lines 311–314 (highlighted in yellow), and to Supplementary Figure 6 a–d. The material and methods have been updated in accordance with the results, pages 25 and 26, lines 689–698 (highlighted in yellow).*

RESULTS

Page 8-9:

To study the importance of L585 in a cellular context, we performed a human endothelial cell-based recycling assay (HERA)⁴³ using human microvascular endothelial cells (HMEC-1) that overexpress human FcRn (HMEC-1-FcRn) (Fig. 3a)⁶⁵. Briefly, adherent HMEC-1-FcRn cells were grown to confluence

before a 1-hour starvation period. Next, equimolar amounts of full-length and truncated QMP were added and incubated for 3 hours to allow for cellular uptake. The cells were then lysed to quantify the amounts of albumin inside or washed and further incubated in serum-free medium for an additional 3 hours to allow for release of recycled albumin back into the medium. The amounts of albumin inside the lysed cells, or recycled albumin in the medium, were quantified by ELISA, in which albumin was captured on an anti-human albumin antibody and detected with an ALP-conjugated anti-human albumin antibody. The results showed that about 5-fold more of both QMP and QMP/LX than WT albumin was taken up (Supplementary Fig. 6 a and c). Moreover, QMP and QMP/LX were recycled equally well, about 8 to 10-fold more efficiently than WT albumin (Fig. 3b, Supplementary Fig. 6e).

Page 12:

First, the FcRn-mediated cellular recycling properties of the full-length and truncated GLP-1 albumin fusions were evaluated by HERA with adherent HMEC-1-FcRn cells, as previously described (Fig. 5a). Following the uptake phase, 1.7-fold more of Tanzeum than GLP-1 WT was detected inside the cells, whereas the amount of GLP-1 LX was similar to that of GLP-1 WT. Both GLP-1 QMP and GLP-1 QMP/LX were detected at about 3-fold higher concentrations than that of GLP-1 WT (Supplementary Fig. 6 b and d).

Supplementary Figure 6. Cellular uptake and recycling of unfused albumin variants and GLP-1 albumin fusions. HERA results showing the amount of **a**, unfused albumin variants and **b**, GLP-1 albumin fusions inside the HMEC-1-FcRn cells following the uptake phase, and relative uptake values for the **c**, unfused albumin variants and **d**, GLP-1 albumin fusions, where WT albumin or GLP-1 WT were set to 1 in each independent experiment. HERA results showing the amount of recycled **e**, unfused albumin variants and **f**, GLP-1 albumin fusions released into the media. The values represent the mean \pm SD from three independent experiments ($n = 3$). Unpaired two-tailed t -tests were used for statistical analysis, where *ns* = not significant, * $p < 0.05$, ** $p < 0.005$, *** $p < 0.0005$, and **** $p < 0.0001$.

MATERIALS AND METHODS

HERA⁴³ was performed by seeding HMEC-1-FcRn cells (1.5×10^5 cells/well) to Costar 48-well TC-treated Multiple Well Plate (Corning) in complete culture medium (250 μ L/well) for approximately 20–24 hours until confluent. The cells were washed three times with pre-warmed Hank's Balanced Salt Solution (HBSS) (Gibco) and starved for 1 hour in HBSS (250 μ L/well). Next, 800 nM of protein samples

(125 $\mu\text{L}/\text{well}$), diluted in HBSS, was added to the cells. After 3 hours of incubation, the cells were washed five times with cold HBSS, and immediately stored at $-20\text{ }^{\circ}\text{C}$ until analysis. A parallel plate treated identically was incubated for an additional 3 hours in fresh assay medium (complete culture medium, without FCS, G418 Sulfate, and Blasticidin S HCl, supplemented with 1X Eagle's MEM Non-Essential Amino Acids Solution (Gibco) ($220\text{ }\mu\text{L}/\text{well}$)). The assay media were collected and stored at $-20\text{ }^{\circ}\text{C}$ until analysis. The plate containing the frozen cells were lysed with Pierce RIPA Lysis and Extraction Buffer (Thermo Scientific) supplemented with 1X cOmplete Protease Inhibitor Cocktail (Roche) ($220\text{ }\mu\text{L}/\text{well}$) for 10 minutes on ice on a tilting board. To remove cell debris, the plates were centrifuged for 10 minutes at 3,500 RPM and $4\text{ }^{\circ}\text{C}$, before the supernatants were collected. The cell lysates (uptake samples) and collected assay media (recycling samples) were analyzed by ELISA.

Reviewer #2 (Remarks to the Author):

Andersen and co-workers describe an approach to use an "engineered" human serum albumin to connect to make it last longer in vivo. Albumin is degraded by carboxypeptidase A (CPA) by removal of L585 that seems to be necessary for binding to FcRn (neonatal Fc receptor). The lifecycle and homeostasis of albumin is dictated by binding to FcRn (after pinocytosis) in intracellular endosomal compartments. If albumin is bound at $\text{pH} < 6.5$ to FcRn, it is trafficked back to the plasma membrane rather than being targeted for degradation in lysosomes.

The main point of the paper is this. The above narrative that they have explored in previous papers can be exploited to create an albumin for attachment to therapeutic peptides and proteins and enhance their half-lives if the attachment does not interfere with the therapeutic peptide's function.

First the authors show through SPR experiments that L585 cannot be substituted away easily by mutagenesis as it also affects binding to FcRn. This was shown by creating 18 variants of albumin replacing L585. Only Ile, Met, and Val substitutions (of L585) resulted in variants that had good affinity for FcRn. But the dilemma is that these are also removed by CPA and diminish the binding to FcRn. Instead they resort to a previously described three-mutation "albumin QMP" (QMP for three of the residues replaced) leaving L585 intact. The salient feature that is being exploited is that somehow even if L585 is lopped off of the QMP-albumin, it retains its ability to bind FcRn at $\text{pHs} < 6.5$ and therefore should retain the recycling back to the plasma membrane and be released. QMP resulted from an earlier investigation from the same group (ref. 62) so that aspect is not novel.

Major point:

The SPR studies are rigorously done and the paper is well written. The in vivo work and affinity data are convincing. So this reviewer is left with the following major concerns about novelty given ref. 62 (Sci. Transl. Med. 2020, 12, eabb0580) from the same group: the "QMP-albumin" has been already described. Addition of GLP-1 to QMP and similar results was certainly one would have expected.

While the construct with GLP-1 attached follows the albiglutide example, pharmacology of GLP-1 at the GLP-1R is entirely missing. The authors must show that this construct that has already gone extensive in vivo lifetime evaluation and also shown to be recycled back to the plasma membrane and released is viable at the GLP-1R at potencies and efficacies rivaling the native peptide.

We thank the reviewer for the suggestion to evaluate the activity of our GLP-1 albumin fusions. To evaluate the ability of the GLP-1 albumin fusions to engage the GLP-1R, we first performed an *in vitro* cell-based assay, where GLP-1 WT, GLP-1 QMP, and unfused WT albumin were incubated with GLP-1R reporter cells expressing a luciferase gene upon activation of the GLP-1R-mediated intracellular pathway. On the following day, luminescence signal was measured. The fusion proteins showed overlapping dose-dependent response curves, resulting in EC₅₀ values of 50.00 nM and 44.41 nM for GLP-1 WT and GLP-1 QMP, respectively. An EC₅₀ value of 0.28 nM was calculated for unfused GLP-1. Thus, the fusion designs are able to activate the GLP-1R at a similar efficacy, but at a lower potency than unfused GLP-1. This is in agreement with previous results reported for albiglutide, and other GLP-1 fusion proteins (Baggio et al., Diabetes, 2004, Bukrinski et al., Biochemistry, 2017, Coskun et al., Mol. Metab., 2018).

To further investigate the ability of GLP-1 albumin fusions to promote insulin secretion in a GLP-1R-dependent manner, we performed an *ex vivo* cell-based assay. Briefly, isolated pancreatic islet cells from WT mice or GLP-1R knock-out mice were cultured overnight, before we incubated the islet cells in a glucose-containing buffer with GLP-1 WT, GLP-1 QMP, or the GLP-1R agonist, Exendin-4. Unfused WT albumin was included as a negative control. Following a 2-hour incubation, the amount of insulin in the supernatants was quantified by ELISA. The results demonstrated that GLP-1 WT, GLP-1 QMP, and Exendin-4 enhanced glucose-dependent insulin secretion from islet cells isolated from the WT mice, as 3 to 4.5-fold higher insulin concentrations were measured than from cells incubated with unfused WT albumin. Furthermore, we found that the enhanced effect of the GLP-1 albumin fusions on insulin secretion was dependent on expression of the GLP-1R, as 2 to 4-fold less insulin was secreted from the islets isolated from the receptor knock-out mice, and no difference compared to the insulin concentrations achieved with unfused WT albumin could be detected. These data are in line with the results from the glucose tolerance test in human FcRn Tg32 mice (Figure 7), which showed that the presence of the GLP-1 albumin fusions increased the plasma concentrations of insulin and reduced glycemic excursion upon challenging the mice with a high dose of glucose. Thus, this collection of data strongly supports that the GLP-1 albumin fusions are functional and able to enhance glucose-dependent insulin secretion through engagement of the GLP-1R expressed by pancreatic islet cells.

The data from the in vitro and ex vivo cell-based assays have been added to the Results, pages 13 and 14, lines 359–379 (highlighted yellow), and to a separate figure (Figure 6), page 38, lines 908-923. The material and methods have been updated in accordance with the results, pages 27 and 28, lines 735–760 (highlighted in yellow). Results from the in vivo GTT experiment are now presented in Figure 7.

RESULTS

GLP-1 albumin fusions enhance insulin secretion through a GLP-1R-dependent mechanism

To test the ability of the GLP-1 albumin fusions to activate the GLP-1R *in vitro*, we used a GLP-1R reporter cell assay kit, where serial dilutions of unfused GLP-1, GLP-1 WT, or GLP-1 QMP were incubated with the GLP-1R reporter cells overnight (Fig. 6a). On the following day, a luciferase detection reagent was added, before luminescence was measured. GLP-1 WT and GLP-1 QMP gave rise to overlapping dose-dependent response curves with EC₅₀ values of 50.00 nM and 44.41 nM, respectively (Fig. 6b). In comparison, a lower EC₅₀ value of 0.28 nM was calculated for unfused GLP-1 (Fig. 6b). Thus, the fusions engaged the GLP-1R expressed by the cells, although at a lower potency than unfused GLP-1, which is in line with previous reports on GLP-1 fusion proteins^{22,74,75}.

Next, we performed an *ex vivo* cell-based glucose-stimulated insulin secretion (GSIS) assay using primary pancreatic islet cells isolated from either WT mice or GLP-1R knock-out mice (Fig. 6c)⁷⁶. When incubated with either GLP-1 WT, GLP-1 QMP, or Exendin-4, the pancreatic islet cells isolated from the WT mice secreted similar amounts of insulin, 3 to 4.5-fold more than when incubated with unfused WT albumin (Fig. 6d). In comparison, the pancreatic islets isolated from the GLP-1R knock-out mice secreted 2 to 4-fold less insulin after incubation with GLP-1 WT, GLP-1 QMP, or Exendin-4, similar amounts to that measured for unfused WT albumin (Fig. 6e). Thus, the results demonstrate that the designed GLP-1 albumin fusions promote glucose-dependent insulin secretion in a GLP-1R-dependent manner. Importantly, the QMP substitutions did not affect the ability of GLP-1 to activate the GLP-1R.

Figure 6. GLP-1 albumin fusions activate the GLP-1R signaling pathway and promote glucose-dependent insulin secretion in pancreatic islet cells. **a**, Illustration of the *in vitro* GLP-1R reporter cell assay. (1) GLP-1R reporter cells were seeded in a 96-well plate and incubated for 5 hours, (2) before the test articles were added. (3) After 23 hours, the luciferase detection reagent was added, and (4) the luminescence was measured after 10 minutes. **b**, Relative luminescence (RLU) measured after incubation with WT albumin (black), GLP-1 WT (purple), GLP-1 QMP (pink), or unfused GLP-1 (blue). The values represent the mean \pm SD from one representative experiment ($n = 3$). The mean EC₅₀ values are shown. **c**, Illustration of the GSIS assay performed with primary pancreatic islets isolated from WT or GLP-1R knock-out mice. (1) Pancreatic islet cells were isolated from the mice and incubated overnight. (2) The islets were incubated in a low-glucose buffer for 2 hours, before the supernatants were collected and (3) replaced with a high-glucose buffer containing the test articles. (4) After 2 hours, the supernatants were collected. (5) ELISA was performed to quantify the secreted insulin concentrations. **d and e**, Concentrations of insulin secreted by islets isolated from (d) WT mice or (e) GLP-1R knock-out mice after incubation with WT albumin (black), GLP-1 WT (purple), GLP-1 QMP (pink), or Exendin-4 (beige). The values represent the mean \pm SD from one experiment ($n = 3-4$), presented as the GSIS values calculated by subtracting the insulin concentrations measured after incubation in the low-glucose buffer from the insulin concentrations measured after incubation in the high-glucose buffer.

MATERIALS AND METHODS

***In vitro* cell-based human GLP-1R reporter assay**

An *in vitro* cell-based assay was performed using a Human GLP-1R Reporter Assay Kit (Indigo Biosciences) according to the manufacturer. Briefly, the GLP-1R reporter cells were seeded into white-bottom 96-well assay plates in cell recovery medium (200 μ L/well) and incubated at 37 °C in a humidified atmosphere of 5% CO₂ for 5 hours. Wells containing medium only were included to quantify the plate-specific fluorescence background signal. The medium was then removed, and serial dilutions of the protein samples (200 μ L/well) were added to the cells (unfused WT albumin, GLP-1 WT, and GLP-1 QMP: 0.08 nM – 6250 nM; GLP-1: 0.00062 nM – 50 nM). After 23 hours of incubation, the supernatants were removed, and a luciferase detection reagent (100 μ L/well) was added. The plate was incubated at room temperature for 5–10 minutes, before luminescence was measured using a Perkin Elmer Victor³ 1420 Multilabel Counter (Wallac 1420 Workstation, version 3.00 revision 5) set at 5 seconds shaking and 0.5 seconds reading.

***Ex vivo* pancreatic islet GSIS assay**

Pancreatic islets were isolated from WT or GLP-1R knockout C57BL/6 mice⁹⁷ using collagenase perfusion, as described⁷⁶. The islets were incubated in RPMI 1640 Medium (Gibco) at 37 °C overnight, before they were washed in Krebs-Ringer HEPES (KRH) solution with 2.8 mM glucose (low-glucose buffer). Five islets per group were incubated at 37 °C for 2 hours in the low-glucose buffer, before the supernatants were collected and stored until analysis. The islets were further incubated in 50 μ L of KRH solution with 16.7 mM glucose (high-glucose buffer), supplemented with 2 μ M of unfused WT albumin, GLP-1 WT, GLP-1 QMP, or exendin-4 (Byetta) (positive control). The islets were incubated at 37 °C for 2 hours, before the supernatants were collected and stored until analysis. Insulin secreted from the islets into the medium was quantified using an insulin ELISA kit (Crystal Chem) according to the manufacturer. GSIS values were calculated by subtracting the insulin concentrations measured after incubation in the low-glucose buffer from the insulin concentrations measured after incubation in the high-glucose buffer.

Minor points:

1. The investigators keep calling methionine a branched amino acid in the manuscript. The methionine side chain has no branching.

We thank the reviewer for pointing this out, and agree that methionine cannot be described as a branched amino acid. Methionine has now been moved to the graph with other non-branched aliphatic amino acids (Fig. 1h), and we have made changes in the text, referring to methionine as a medium-long aliphatic amino acid, or by its name (*highlighted in yellow on pages 2, 5, 7, 18, and 32*).

2. Page 4, lines 104-106: this sentence needs to be rewritten. The N-terminus of GLP-1 is indeed necessary for activity at the GLP-1R but the way this sentence is written, it seems that the N-terminus of albumin is attached to GLP-1 -- not sure how this makes sense.

We apologize for the confusion regarding the protein design. GLP-1 requires a free N-terminal end for optimal binding to its GLP-1R. The GLP-1 albumin fusions contain two copies of the GLP-1 peptide in tandem, where the C-terminal end of the second copy is genetically fused to the N-terminal end of albumin (fusion protein written from N' to C': [N' GLP-1 C']-[N' GLP-1 C']-[N' albumin C']) (Fig. 4a), leaving the N-terminal end of the first copy of GLP-1 free to interact with the GLP-1R.

To clarify, we have modified the sentence in the manuscript (*page 4, lines 105–106, highlighted in yellow*):

“As GLP-1 in albiglutide requires an available N-terminal end for engagement with the GLP-1 receptor (GLP-1R)^{61,62}, albumin is fused to the C-terminal end of the peptide. However, such design leaves the C-terminal end of albumin free and susceptible to enzymatic cleavage by CPA.”

3. Page 6, Lines 130-131: "CPA cleaves L585, but not the following glycine residue (G584)". Protease cleave in between residues. They don't cleave any individual "residues". The authors should state "that CPA catalyzes hydrolysis of the amide bond between L585 and ..."

We thank the reviewer for pointing out this important distinction in describing enzymatic cleavage. We agree, and have now used the suggested sentence (*page 6, lines 131–133, highlighted in yellow*): “While CPA catalyzes hydrolysis of the amide bond between L585 and G584, it does not of the following amide bond between G584 and L583.”

Overall, this is a good paper and deserves publication with concerns about novelty and a very major concern experimentally about showing GLP-1 activity.

Reviewer #3 (Remarks to the Author):

It this article authors have presented progress on their work where they had identified the C-terminal amino acid of albumin important for its tight binding to FcRn and QMP mutation as beneficial in terms of increasing FcRn binding. The authors show in this paper that QMP can overcome the deletion of c-terminal leucine, and substitution of leucine with other amino acids that have branched aliphatic chain does not affect FcRn binding, They also show that fusion of QMP albumin with modified GLP-1 peptide retains the higher FcRn binding despite c-terminal leucine deletion. Lastly, PK and PD of GLP-1+albumin fused protein is shown to further support the conclusion. In general, the paper is well written, and only minor revisions are needed. Below are a few comments for the authors to address.

1. While the authors have briefly mentioned that GLP-1 fused albumin construct had shorter half-life that just albumin, there is not mechanistic explanation provided for it. It will be good to show maybe FcRn binding data or some other properties of the molecules that leads to this difference in pharmacokinetic behavior.

We agree with the reviewer that this is an observation that is important to address. The half-lives of the GLP-1 albumin fusions were about 1.8-fold shorter than that of unfused albumin in the human FcRn Tg32 mice. This is in agreement with the PK data for Tanzeum in humans, for which a half-life of 5 days on average has been reported, about 4-fold shorter than that of endogenous albumin (20 days).

Due to the high plasma concentrations of endogenous albumin (40 mg/mL), the competitive pressure on the albumin binding site of FcRn is substantial. Thus, even minor alterations in receptor binding of a fusion product may alter the ability of the receptor to rescue it from intracellular degradation. Unexpectedly, when we measured the FcRn binding kinetics at pH 5.5 by SPR, where we injected the unfused albumin variants or the GLP-1 albumin fusions over immobilized recombinant soluble human FcRn, we found that the GLP-1 albumin fusions exhibited about 5-fold stronger binding affinities

towards the receptor than the corresponding unfused albumin variants. However, when we measured FcRn-mediated recycling in HMEC-1-FcRn cells using HERA, we observed less efficient recycling of the GLP-1 albumin fusions than of the corresponding unfused albumin variants, which explains the shorter plasma half-lives measured for the GLP-1 albumin fusions *in vivo*.

Interestingly, it has previously been demonstrated that the biophysical properties of IgG antibodies affect the cellular handling and half-life of such molecules (references 81-85 in the manuscript). Indeed, IgG antibodies demonstrating equal FcRn binding properties in biochemical assays can have very different plasma half-lives, due to differences in biophysical properties, including pI and net charge, which affect how they are handled in a cellular context. For the GLP-1 albumin fusions, we calculated the same theoretical pI as for unfused albumin, and only small differences in pH-dependent net charge. Thus, this probably does not explain the discrepancy between the *in vitro* FcRn binding and cellular handling or *in vivo* half-life.

Another possible explanation for the difference in PK behavior may be because the GLP-1 albumin fusions engage their target, the GLP-1R, *in vivo*. The results from the glucose tolerance test performed in Tg32 mice (Figure 7), showing that the GLP-1 albumin fusions increase plasma insulin concentrations and limit glycemic excursion, support that the fusions engage the mouse GLP-1R. This is further supported by new data from an *ex vivo* cellular assay, where we incubated pancreatic islets cells, isolated from either WT mice or GLP-1R knock-out mice, with the GLP-1 albumin fusions (Figure 6). The results demonstrated that the fusions enhanced insulin secretion in a GLP-1R-dependent manner. Thus, engagement of the GLP-1R *in vivo* may contribute to faster clearance of the fusion proteins than albumin alone.

Importantly, the half-life of the GLP-1 WT fusion protein in our study is much longer than that of the unfused peptide (~2 hours of the DPP-4 resistant Exenatide; Knop et al., *Exp Opin Pharmacother*, 2017), and QMP engineering further improves the PK properties of the fusion protein, such that more may reach its target.

We have now included this topic in the discussion, please refer to page 16-17, lines 423–425 and lines 428–439.

Consistent with the human PK data, we observed that the half-life of GLP-1 WT in transgenic mice expressing human FcRn was 1.8-fold shorter than that of unfused WT albumin. **Engagement of its target receptor, the GLP-1R, is likely one factor contributing to the faster clearance rate for the GLP-1 albumin fusion.** Moreover, given the high plasma concentrations of endogenous albumin (40 mg/mL), the competitive pressure on the albumin binding site of FcRn is substantial⁶³. Inevitably, even minor alterations in FcRn binding properties of a protein fusion can therefore affect the ability of the receptor to rescue it from intracellular lysosomal degradation. **Unexpectedly, we found that GLP-1 WT bound 5-fold more strongly than unfused WT albumin to human FcRn at acidic pH, due to both faster association and slower dissociation. In this case, the binding was measured towards a recombinant truncated and soluble form of the receptor. When we evaluated FcRn-mediated recycling in a cell-based assay, however, where the receptor is embedded in the membrane, the WT fusion was recycled 1.8-fold less efficiently than unfused WT albumin, which is in line with its shorter plasma half-life. In this regard, it is interesting that IgG antibodies that show similar FcRn binding properties can have very different plasma half-lives due to differences in biophysical properties, such as pI and net charge, which affect cellular handling⁸¹⁻⁸⁵. However, as the GLP-1 albumin fusion and unfused albumin have the same theoretical pI, and only small differences in pH-dependent net charge, this probably does not explain the discrepancy between their FcRn binding properties and cellular handling or plasma half-life.**

Calculations of the isoelectric point (pI) and pH-dependent net charge of unfused albumin variants and GLP-1 albumin fusions have been added to the Results, page 8, lines 201–204, and page 10, lines 253–255, Supplementary Table 3, and Supplementary Fig. 5. The material and methods have been updated in accordance with the results, pages 23 and 24, lines 628–631 (highlighted in yellow).

RESULTS

Page 8:

Lastly, the theoretical net charge of the proteins was determined as a function of pH (pH 1.0 to pH 13.9) (Supplementary Fig. 5a). An isoelectric point (pI) of 5.8, and net charges of +2.8 and –14.7 at pH 5.5 and pH 7.4, respectively, were calculated for the albumin variants (Supplementary Table 3).

Page 10:

A theoretical pI of 5.8, and net charges of +2.9 and –15.5 at pH 5.5 and pH 7.4, respectively, were calculated for the GLP-1 albumin fusions (Supplementary Fig. 5b, Supplementary Table 3).

Supplementary Table 3. Theoretical pI and net charge of unfused albumin variants and GLP-1 albumin fusions

Protein	pI ^a	Net charge	
		pH 5.5	pH 7.4
WT	5.8	+2.8	–14.7
QMP			
QMP/LX			
(GLP-1 (7-36)) ₂	5.6	+0.1	–2.22
GLP-1 WT	5.8	+2.9	–15.5
GLP-1 LX			
GLP-1 QMP			
GLP-1 QMP/LX			

^a pI, isoelectric point.

Supplementary Figure 5. pH-dependent net charge of unfused albumin variants and GLP-1 albumin fusions. Theoretical net charge of **a**, unfused albumin variants and **b**, unfused GLP-1 or GLP-1 albumin fusions, as a function of pH (pH 1 to pH 13.9), calculated with EMBOSS iep.

MATERIALS AND METHODS

Prediction of pI and net charge

The EMBOSS iep Software (EMBOSS) (https://www.bioinformatics.nl/cgi-bin/emboss/iep?_pref_hide_optional=0) was used to calculate the theoretical pI, and net charge at different pH values of unfused albumin variants and GLP-1 albumin fusions.

2. Please discuss what will happen if the proteins such as GLP-1 are fused to the C-terminal? Would they be cleaved in the presence of CPA, or will create steric hindrance for CPA and prevent c-terminal deletion.

Knowing that the C-terminal end of albumin is free and susceptible for CPA cleavage in fusion designs where the therapeutic agent is fused to its N-terminal end, one may consider whether the opposite orientation would be preferable to prevent enzymatic cleavage. As CPA is a carboxypeptidase that cleaves peptide bonds from the C-terminal end of proteins, albumin would not be accessible as a substrate for the enzyme if the therapeutic agent was fused to its C-terminal end. Thus, fusion to the C-terminal end of albumin could be an attractive alternative, as long as that the therapeutic agent itself is not a substrate for CPA and this orientation does not compromise FcRn engagement.

CPA has affinity for protein substrates with a branched-chained or aromatic C-terminal amino acid residue. In regard to GLP-1 in our designs, and in albiglutide (Tanzeum), the C-terminal amino acid is arginine, which is not recognized by CPA, and thus would not be removed by the enzyme if encountered upon *in vivo*. However, as the N-terminal end of GLP-1 is required for engagement of the GLP-1R, fusion to the C-terminal end of albumin would compromise its activity. Thus, this orientation is not applicable for the GLP-1 peptide, but may certainly be for other therapeutic agents.

We have previously studied whether genetic fusion of a peptide or a single-chain variable fragment to either the N-terminal or C-terminal end of albumin affects FcRn binding (Andersen et al., J Biol Chem, 2013). The most prominent effect was observed upon C-terminal fusion, which reduced the FcRn binding affinity by up to 2-fold. This may relate to the fact that the C-terminal domain III of albumin contains the main binding site for FcRn (Andersen et al., Nat. Commun., 2013), and that the last C-terminal α -helix requires a certain degree of flexibility to engage the receptor optimally (Nilsen et al., Commun Biol., 2020).

Nonetheless, the orientation is worth considering when designing new albumin fusions, and regardless of the choice, FcRn binding and receptor-mediated cellular recycling should be addressed for each fusion, as the nature of the therapeutic agent may interfere with these processes at varying degrees.

We have now included this topic in the discussion, please refer to page 17, lines 452–461 (highlighted in yellow).

Although not applicable for GLP-1, fusion to the C-terminal end of albumin could be an attractive approach to avoid cleavage by CPA. However, this strategy is only attractive if the therapeutic agent itself is not a substrate for CPA, or if the case, that its functional activity is not negatively affected by C-terminal cleavage. Moreover, it is important that FcRn engagement is not hampered upon fusion to the C-terminal DIII of albumin, which contains the main binding site for the receptor^{47,53}. We previously found that fusion of a peptide or a single-chain variable fragment to the C-terminal end of albumin

reduced FcRn binding up to 2-fold, and to a greater extent than when fused to the N-terminal end⁶⁷. However, this may vary depending on the nature of the fused biologic. Therefore, the orientation should be considered on a case-to-case basis upon designing new albumin fusions.

3. The conclusion that QMP provides better subcutaneous bioavailability seems incorrect based on figure 5g and supplementary table 3. After IV administration the ratio of QMP and WT albumin AUC is ~3.8, and that for QMP and WT fused proteins is ~2.7. After SC administration the ratio for QMP and WT fused protein AUCs is ~3.1. Thus, it does not seem bioavailability is changing much, and the apparent difference in AUC mainly stems from different PK. While higher binding to FcRn at pH 6 has been shown to provide better subcutaneous bioavailability of antibody, the data in this case does not seem convincing enough.

We thank the reviewer for pointing this out. What we aimed to communicate is that the GLP-1 albumin fusions show high bioavailability following SC injection. Each GLP-1 albumin fusion gave rise to similar AUC values following IV and SC administration, and as pointed out by the reviewer, the QMP engineering increased the area under the time-concentration curve by about 3-fold, irrespective of administration route. This supports that the GLP-1 albumin fusions can be administered subcutaneously, which is the most common route of administration for GLP-1R agonists used in the clinic. However, we agree that we cannot claim that QMP engineering increases the bioavailability of the fusion based on the data, and have removed or adjusted text accordingly.

4. Discussion is just repetition of results and can benefit from broader perspective of authors on implications of this work and its comparison with findings from other groups on similar subject matter.

We thank the reviewer for the suggestion to expand our perspective in the discussion, and for the questions asked, which inspired us to add the following new sections:

- Pages 16–17, lines 428–439: contributing factors to different PK properties of unfused albumin and albumin fusions.

Consistent with the human PK data, we observed that the half-life of GLP-1 WT in transgenic mice expressing human FcRn was 1.8-fold shorter than that of unfused WT albumin. Engagement of its target receptor, the GLP-1R, is likely one factor contributing to the faster clearance rate for the GLP-1 albumin fusion. Moreover, given the high plasma concentrations of endogenous albumin (40 mg/mL), the competitive pressure on the albumin binding site of FcRn is substantial⁶³. Inevitably, even minor alterations in FcRn binding properties of a protein fusion can therefore affect the ability of the receptor to rescue it from intracellular lysosomal degradation. Unexpectedly, we found that GLP-1 WT bound 5-fold more strongly than unfused WT albumin to human FcRn at acidic pH, due to both faster association and slower dissociation. In this case, the binding was measured towards a recombinant truncated and soluble form of the receptor. When we evaluated FcRn-mediated recycling in a cell-based assay, however, where the receptor is embedded in the membrane, the WT fusion was recycled 1.8-fold less efficiently than unfused WT albumin, which is in line with its shorter plasma half-life. In this regard, it is interesting that IgG antibodies that show similar FcRn binding properties can have very different plasma half-lives due to differences in biophysical properties, such as pI and net charge, which affect cellular handling⁸¹⁻⁸⁵. However, as the GLP-1 albumin fusion and unfused albumin have the same theoretical pI, and only small differences in pH-dependent net charge, this probably does

not explain the discrepancy between their FcRn binding properties and cellular handling or plasma half-life.

- Page 17, lines 452–461: orientation of fusion protein design related to CPA cleavage.

Although not applicable for GLP-1, fusion to the C-terminal end of albumin could be an attractive approach to avoid cleavage by CPA. However, this strategy is only attractive if the therapeutic agent itself is not a substrate for CPA, or if the case, that its functional activity is not negatively affected by C-terminal cleavage. Moreover, it is important that FcRn engagement is not hampered upon fusion to the C-terminal DIII of albumin, which contains the main binding site for the receptor^{47,53}. We previously found that fusion of a peptide or a single-chain variable fragment to the C-terminal end of albumin reduced FcRn binding up to 2-fold, and to a greater extent than when fused to the N-terminal end⁶⁷. However, this may vary depending on the nature of the fused biologic. Therefore, the orientation should be considered on a case-to-case basis upon designing new albumin fusions.

- Page 18, lines 488–497: alternative half-life extension strategy

One alternative approach to direct fusion to albumin, in which one would not have the risk of C-terminal end cleavage limiting the therapeutic potential, is by fusing or conjugating GLP-1 or other therapeutic agents to an albumin-binding molecule⁷. This strategy has been exploited, for instance, in the long-acting GLP-1RA semaglutide (Novo Nordisk), which was approved for treatment of T2DM in 2017, and for treatment of obesity in 2021, and has since then become the most prescribed GLP-1RA⁸⁷. In semaglutide, DPP-4 resistant GLP-1 is conjugated to a fatty acid chain of 18 carbon atoms, a natural ligand of albumin⁸⁸. The drug therefore non-covalently associates with endogenous albumin upon administration, which extends the half-life of the peptide from a few hours to 7 days, enabling once-weekly injections^{88,89}. However, it also means that the strategy is limited by the half-life of endogenous WT albumin.

References:

- Andersen, J. T., et al. (2012). "Structure-based mutagenesis reveals the albumin-binding site of the neonatal Fc receptor." *Nat. Commun.* **3**, 610.
- Andersen, J. T. et al. (2013). "Single-chain variable fragment albumin fusions bind the neonatal Fc receptor (FcRn) in a species-dependent manner: implications for in vivo half-life evaluation of albumin fusion therapeutics." *J. Biol. Chem.* **288**, 24277–24285.
- Andersen, J. T., et al. (2014). "Extending serum half-life of albumin by engineering neonatal Fc receptor (FcRn) binding." *J. Biol. Chem.* **289**(19): 13492-13502.
- Baggio et al., (2004). "A recombinant human glucagon-like peptide (GLP)-1-albumin protein (Albugon) mimics peptidergic activation of GLP-1 receptor-dependent pathways coupled with satiety, gastrointestinal motility, and glucose homeostasis." *Diabetes* **53**, 2492-2500.
- Benjakul, S., et al. (2023). "A pan-SARS-CoV-2-specific soluble angiotensin-converting enzyme 2-albumin fusion engineered for enhanced plasma half-life and needle-free mucosal delivery." *PNAS Nexus*.
- Bukrinski, J. T. et al. (2017). "Glucagon-like Peptide 1 Conjugated to Recombinant Human Serum Albumin Variants with Modified Neonatal Fc Receptor Binding Properties. Impact on Molecular Structure and Half-Life." *Biochemistry* **56**, 4860-4870.
- Bern, M., et al. (2020). "An engineered human albumin enhances half-life and transmucosal delivery when fused to protein-based biologics." *Science translational medicine* **12**(565): eabb0580.
- Coskun, T. et al. (2018) "LY3298176, a novel dual GIP and GLP-1 receptor agonist for the treatment of type 2 diabetes mellitus: From discovery to clinical proof of concept." *Mol Metab* **18**, 3-14.
- Knop, F. K., et. al. (2017). Exenatide: pharmacokinetics, clinical use, and future directions. *Expert Opinion on Pharmacotherapy*, **18**(6), 555–571.
- Lombardi S., et al. (2021). "Fusion of engineered albumin with factor IX Padua extends half-life and improves coagulant activity." *British Journal of Haematology*, **194**(2): 453-462.
- Nilsen, J., et al. (2020) "An intact C-terminal end of albumin is required for its long half-life in humans". *Commun. Biol.* **3**, 181.

Response to referee 2:

We acknowledge the reviewer's attention to the potency of the GLP-1 albumin fusions and longevity of their therapeutic effect.

Based on how we interpret the response, the reviewer is not convinced that the GLP-1 QMP fusion will have better therapeutic efficacy compared to the commercial product albiglutide. The reviewer questions whether the GLP-1 QMP fusion is therapeutically relevant, because of the 157-fold lower potency exhibited compared to the native human GLP-1 peptide, as determined using an GLP-1R reporter cell assay (Fig. 6b). In this regard, the reviewer refers to a 100-fold loss of potency for albiglutide, presumably taken from Baggio et al. (Diabetes, 2004). Importantly, a different experimental approach was taken in this published paper, where baby hamster kidney cells, transiently transfected to express the rat GLP-1R, was used, and albiglutide was compared with Exendin-4, a homolog isolated from *Heloderma suspectum*, which has 53% homology to human GLP-1. Thus, caution should be taken when directly comparing these results to our study, as we have used a reporter cell line expressing the human form of GLP-1R to compare GLP-1 WT and GLP-1 QMP with that of the native human GLP-1 peptide.

We would like to emphasize that our designed GLP-1 WT fusion (containing wild-type human albumin) is a recombinant version of the product albiglutide/Tanzeum. Hence, they have identical amino acid sequences. In line with this, we show that GLP-1 WT and albiglutide have similar human FcRn binding and cellular recycling properties, and they show the same plasma half-life in human FcRn transgenic mice (Supplementary Fig. S8). In the same mice, we also show that the insulin stimulating- and glucose-lowering properties of GLP-1 WT is similar to that of albiglutide/Tanzeum (Fig. 7b). Therefore, we used the GLP-1 WT fusion as an "albiglutide-mimic" in the other experiments where we compared the functionality with that of our human albumin engineered version with improved human FcRn binding, namely GLP-1 QMP. This was needed as we had very limited amounts of albiglutide available as it is no longer obtainable from the pharmacy. Notably, albiglutide/Tanzeum was on the market from 2014 to 2018, approved for treatment of type 2 diabetes (Poole et al., Drugs, 2014), before GlaxoSmithKline discontinued the product due to financial priorities in a highly competitive market.

Specifically, using the GLP-1R reporter cell assay, we found that the GLP-1 WT (EC₅₀= 50.0 nM) and GLP-1 QMP (EC₅₀= 44.4 nM) fusions engage the GLP-1R equally well, and they both demonstrate lower potency, 178- and 157-fold, respectively, compared to the native human GLP-1 peptide (EC₅₀= 0.28 nM) (Fig. 6b). These results support that the GLP-1 QMP fusion exhibits the same potency as the GLP-1 WT albiglutide-mimic, and therefore the same as albiglutide.

Furthermore, albiglutide has been reported to have a plasma half-life of 5 days on average in humans, which allowed for once-weekly administration (ref. 25-27 in manuscript). In a phase 2 clinical study (Rosenstock et al., Diabetes Care, 2009), albiglutide was compared to twice-daily dosing of Exendin-4 (half-life of 2.4 h), which was included as an open-label reference. Once-weekly dosing of albiglutide for 16 weeks resulted in a reduction in glycosylated hemoglobin of 0.87%, whereas a reduction of 0.54% was measured for the group receiving Exendin-4 (Byetta) twice-daily. Additionally, a 1.44% reduction in fasting plasma glucose was achieved with albiglutide, which was greater compared to a 0.80% reduction reported for Exendin-4. Thus, despite the reduction in potency, albiglutide proved to be effective in improving glycemic control in patients with type 2 diabetes. The study also highlights the fact that the plasma half-life of the therapeutic drug dictates the dosing regimen.

Another key feature is that the plasma half-life of albiglutide (5 days) in humans is much shorter than that of endogenous wild-type human albumin (20 days) (ref. 25-27 in manuscript). Moreover, as the GLP-1 di-peptide is fused to the N-terminal end of albumin, the fusion can be cleaved by CPA at the C-terminal end. Removal of the last leucine residue (L585) of wild-type human albumin weakens the binding to human FcRn, which we have shown to reduce the plasma half-life from 20 days to 3.5 days for albumin in humans. This may have great therapeutic implications for the plasma concentration of a wild-type albumin fusion over time. Thus, the main aim of our manuscript is to report on a strategy that can secure favorable FcRn binding and pharmacokinetic properties of human albumin-fused therapeutic candidates. As such, we present data that support QMP engineering of albumin as an advantageous approach, where three amino acid substitutions, which greatly improves pH-dependent human FcRn binding, are compensating for the reduction in binding caused by removal of L585 in the context of fusion modalities. While removal of the L585 residue greatly decreased FcRn binding of fusion designs containing wild-type albumin, this was not the case for the corresponding QMP-engineered variants. Consequently, a shorter plasma half-life was measured for the truncated GLP-1 WT lacking L585 (GLP-1 L585X) in human FcRn expressing mice, whereas the GLP-1 QMP fusion demonstrated favorable FcRn binding and a long plasma half-life of about 3 days, irrespective of the presence of L585.

To study whether the extended plasma half-life of the GLP-1 QMP fusion would increase the longevity of the therapeutic effect, we repeated the glucose tolerance test in human FcRn expressing mice with more time, six hours instead of one hour, between the dosing of the GLP-1 fusions and injection of the high dose of glucose. We here compared GLP-1 QMP with the recombinant albiglutide-mimic, GLP-1 WT. The results showed that GLP-1 QMP more efficiently lowered the blood glucose compared with GLP-1 WT, as 2-fold lower glucose levels were measured ten minutes after the glucose injection in the mice that had received the QMP-engineered fusion (Fig. 7d-f). These results demonstrate that QMP engineering prolongs the blood glucose lowering effect, which is likely a result of the extended plasma half-life, and thus, a higher plasma concentration of GLP-1 QMP than of GLP-1 WT at the time of the glucose challenge.

Thus, we present data supporting that the GLP-1 QMP fusion exhibits an extended plasma half-life as well as a prolonged therapeutic effect compared with the same fusion design containing wild-type human albumin. In addition, we show that the C-terminal end of GLP-1 QMP can be enzymatically cleaved without affecting plasma-half-life.

To further clarify that the designed GLP-1 WT fusion is a recombinant version of albiglutide/an albiglutide-mimic, we have modified some of the text in the results (highlighted in yellow on lines 121-122, 240, 245, 258, 302-303, 326-327, 367, 369, 394-395 and 400).

(4) Now there are new issues: Why would one expect enhanced insulin secretion with a fused albumin? "RESULTS" in response to reviewer 2.

We are sorry about the confusion. As part of its natural functions, GLP-1 enhances insulin secretion from pancreatic beta cells in the presence of glucose. In the new section of the results, we do not claim that the GLP-1 albumin fusions enhance glucose dependent insulin secretion beyond that of the unfused peptide, rather that the fusions do so to a similar degree. This is supported by the new data obtained using the ex vivo cell-based glucose stimulated insulin secretion (GSIS) assay with primary pancreatic islet cells isolated from WT mice (Fig. 6d). When incubating the islet cells with either GLP-1 WT, GLP-1 QMP or Exendin-4 in the presence of glucose, 3 to 4.5-fold more insulin was

secreted compared to when incubated with unfused WT albumin (negative control protein) (Fig. 6d). As such, to avoid any confusion, we have replaced “enhance or promote” with “stimulate” in the results section (highlighted in yellow on lines 360, 378 and 911).

References:

Baggio et al., (2004). “A recombinant human glucagon-like peptide (GLP)-1-albumin protein (Albugon) mimics peptidergic activation of GLP-1 receptor-dependent pathways coupled with satiety, gastrointestinal motility, and glucose homeostasis.” *Diabetes* **53**, 2492-2500.

Rosenstock et al., (2009). “Potential of albiglutide, a long-acting GLP-1 receptor agonist, in type 2 diabetes: a randomized controlled trial exploring weekly, biweekly, and monthly dosing.” *Diabetes Care*, 32(10):1880-6.

Poole et al., (2014). “Albiglutide: first global approval.” *Drugs* **74**, 929-938.